

# Version 8 IMK/IAA MIPAS measurements of ClO

Norbert Glatthor[1], Thomas von Clarmann[1,†], Udo Grabowski[1], Sylvia Kellmann[1], Michael Kiefer[1], Alexandra Laeng[1], Andrea Linden[1,†], Gabriele P. Stiller[1], Bernd Funke[2], Maya García-Comas[2], Manuel López-Puertas[2], Oliver Kirner[3], and Michelle L. Santee[4]

[1]Karlsruhe Institute of Technology, Institute of Meteorology and Climate Research, Karlsruhe, Germany
[2]Instituto de Astrofísica de Andalucía, CSIC, Granada, Spain
[3]Karlsruhe Institute of Technology, Scientific Computing Center, Karlsruhe, Germany
[4]Jet Propulsion Laboratory, California Institute of Technology, Pasadena, California, USA
[†]deceased, Thomas von Clarmann on 13 January 2024 and Andrea Linden on 18 November 2024

**Correspondence:** Norbert Glatthor (norbert.glatthor@kit.edu)

**Abstract.** Global distributions of chlorine monoxide (ClO) were retrieved from infrared limb emission spectra recorded with the Michelson Interferometer for Passive Atmospheric Sounding (MIPAS), covering the time period from July 2002 to April 2012. The retrieval was performed by constrained non-linear least squares fitting using spectral lines in the fundamental band of ClO around 844 $cm^{-1}$. The vertical resolution of V8 ClO is 4 km at 18–20 km and 7.5–9.5 km at 40 km altitude. The

considerable improvement at 40 km with respect to the previous V5 data version is achieved by extension of the spectral range for retrieval of upper stratospheric ClO. Errors are by far dominated by measurement noise and increase from 0.4–0.5 ppbv at 20 km to 0.8 ppbv at 50 km altitude. Thus, in general, individual ClO profiles are noisy, and profile averaging has to be performed for, e.g., analysis of the upper stratospheric maximum. However, strongly enhanced lower stratospheric ClO amounts of more than 1.5 ppbv during polar winter are well detected in single measurements. Along with the standard representation of the data,

an alternative coarse grid representation that obviates the need to apply averaging kernels in certain situations is also provided. Due to improved modeling of the atmospheric continuum and the instrumental offset, the high bias in upper stratospheric ClO that had particularly affected the previous V5 data over the period 2005–2012 has been removed. A comparison with ClO measurements of the Microwave Limb Sounder (MLS) on the Aura satellite shows good agreement between the lower stratospheric enhancements observed by the two instruments in polar winter. There is also good agreement between the upper

stratospheric ClO amounts observed in the northern hemisphere and at southern hemispheric low latitudes. With the support of simulations from the Earth system model ECHAM/MESSy Atmospheric Chemistry (EMAC), deviations between the ClO amounts of MIPAS and MLS in the Antarctic lower stratosphere during July and in the upper stratosphere, especially at southern mid- and high latitudes during winter, are attributed to the different local solar times of the measurements.

## 1 Introduction

Since Molina and Rowland (1974) predicted stratospheric ozone destruction by a catalytic cycle involving Cl and ClO, measurements of reactive chlorine have become a key activity in atmospheric science. The discovery of the Antarctic ozone hole (Chubachi, 1984; Farman et al., 1985) came as a surprise because main ozone destruction was expected in the tropical strato-



sphere. Solomon et al. (1986) could solve this puzzle by postulating a mechanism that involved heterogeneous reactions of the reservoir species HCl and $ClONO_2$ on the surface of polar stratospheric cloud particles, which ultimately leads to the release of reactive chlorine species like ClO. Even under polar spring conditions, when more sunlight is available, ozone destruction via the catalytic ClO cycle is only a minor pathway for ozone destruction in the polar lower stratosphere (Salawitch et al., 1993).

However, during nighttime substantial amounts of ClO are transformed into its dimer ClOOCl, and the catalytic ClO dimer cycle, which does not involve atomic oxygen, is the chief reaction chain responsible for the polar ozone hole (Molina and Molina, 1987; Barrett et al., 1988; Cox and Hayman, 1988; Anderson et al., 1989). A more detailed discussion of stratospheric chlorine chemistry and the history of its discovery is given in von Clarmann (2013).

After the discovery of the Antarctic ozone hole, a global agreement, the Montreal Protocol, was adopted in 1987 to

limit emissions of ozone depleting substances in order to protect the ozone layer (https://www.unep.org/ozonaction/who-we-are/about-montreal-protocol). This agreement was followed by various amendments. However, evidence of stratospheric ozone recovery is just beginning to emerge (e.g. World Meteorological Organization (WMO), 2022, and references therein), and the question if latitudinally and vertically resolved stratospheric ozone trends are explainable by corresponding trends in stratospheric chlorine, in particular ClO, is still waiting for its final answer.

The record of stratospheric ClO measurements is quite extensive. The first observational confirmation of the existence of large amounts of ClO in the stratosphere was made by balloon-borne in situ measurements, using resonance fluorescence (Anderson et al., 1977, 1980). Menzies (1979) used balloon-borne infrared solar absorption measurements by a heterodyne radiometer, while Waters et al. (1981) relied on limb observations in the microwave range. Shortly after, ground-based microwave measurements were performed by Parrish et al. (1981). Microwave limb emission spectroscopy was also used for the

first global space-borne ClO measurements by the Microwave Limb Sounder on the Upper Atmospheric Research Satellite (Waters et al., 1993). In the following years, further ClO measurements from satellite platforms were made by the Millimeter-wave Atmospheric Sounder (MAS) on the Space Shuttle (Aellig et al., 1996), the Michelson Interferometer for Passive Atmospheric Sounding (MIPAS) on Envisat (Glatthor et al., 2004; von Clarmann et al., 2009b), the Atmosphere Chemistry Experiment-Fourier Transform Spectrometer (ACE-FTS) on SciSat-1 (Dufour et al., 2006), the Sub-Millimeter Radiometer (SMR) on

Odin (Urban et al., 2004, 2005), the Microwave Limb Sounder (MLS) on the Aura satellite (Santee et al., 2008) and the Superconducting Submillimeter-Wave Limb-Emission Sounder (SMILES) on the International Space Station (ISS) (Sato et al., 2012).

These days, monitoring of stratospheric ClO from the ground is a routine activity, in particular by stations associated with the Network for Detection of Atmospheric Composition Change (NDACC) (e.g. Solomon et al., 1984; de Zafra et al., 1994;

Raffalski et al., 1998; Solomon et al., 2000; Nedoluha et al., 2011, 2025). These measurements were complemented by observations within the framework of specific measurement campaigns, using ground-based (de Zafra et al., 1989), airborne (e.g. Crewell et al., 1994; Wehr et al., 1995) and balloon-borne (e.g. Stachnik et al., 1992, 1999; Wetzel et al., 2010; de Lange et al., 2012) platforms.

The focus of this paper is on updates of the retrieval of ClO volume mixing ratios (VMRs) from MIPAS with the level-2

research data processor developed and operated by the Institute for Meteorology and Climate Research (IMK) in cooperation



with the Institute de Astrofísica de Andalucía (IAA-CSIC), and on the presentation of a MIPAS ClO climatology and a comparison with MLS measurements. After a short description of the MIPAS instrument and its two operational phases (Section 2), we review unresolved issues and problems with preceding data versions (Section 3). In Sections 4, 5 and 6 we present the mid-infrared signature of ClO, describe our actions performed to solve the deficiencies in preceding data versions and perform

a significance test of the ClO retrievals. In Section 7 we turn towards a data characterization including averaging kernels, the horizontal distribution of information, presentation of the ClO VMRs, the corresponding estimated standard deviation (ESD) and vertical resolution along selected orbits, followed by a discussion of the error budget. For data users who prefer not to work with averaging kernels, we also provide the data on a coarse grid, where averaging kernels do not need to be applied. In Section 8 we compare time series of monthly zonal means of V8 ClO with the respective V5 data and present a climatology

of V8 ClO, followed by a comparison with MLS data in Section 9. To investigate the reason for differences between MIPAS and MLS observations in the upper stratosphere, simulations from the European Centre/Hamburg/Modular Earth Submodel System Atmospheric Chemistry (EMAC) model (Jöckel et al., 2006, 2010) are also taken into account. Finally, we summarize the results and the improvements achieved with the new ClO data set (Section 10).

## 2  MIPAS, MLS and EMAC

The MIPAS instrument was a Fourier transform limb emission spectrometer operated on the Envisat platform at 800 km altitude (Fischer et al., 2008). MIPAS observed the atmosphere over an altitude range from the upper troposphere to the mesosphere – and occasionally the lower thermosphere – from a polar sunsynchronous orbit of 98.55° inclination with an Equator crossing at 10:15 AM local solar time on the descending part of the orbit. MIPAS was employed with its full spectral resolution of 0.025 cm$^{-1}$ (spectral sampling), corresponding to a spectral resolution of 0.05 cm$^{-1}$ after apodization with the "strong" apodization

function given by Norton and Beer (1976), from June 2002 to April 2004 (full resolution (FR) period). A technical defect of the interferometer slide forced operation at a coarser spectral resolution of 0.0625 cm$^{-1}$ (0.12 cm$^{-1}$ after apodization) from January 2005 onwards until the end of the mission in April 2012 (reduced resolution (RR) period). The sampling rate of MIPAS was 1070 scans per day during the FR period and 1400 scans per day during nominal mode observations in the RR period.

For this study, we use spectra version 8.03 provided by the European Space Agency. Details of the calibration and the char-

acterization of the spectra are provided by Kleinert et al. (2018) and Kiefer et al. (2021). The measurement modes of MIPAS utilized for ClO evaluation are the NOminal Mode (NOM, tangent altitudes from about 6 to 70 km), the Upper Troposphere Lower Stratosphere mode (UTLS-1, 5.5 – 55 km), and the Middle Atmosphere mode (MA, 18 – 102 km) (Oelhaf, 2008). The latter two modes were applied during the RR period only. The different data versions of V8 ClO are summarized in Table 1.

MLS is one of four instruments on the Earth Observing System (EOS) Aura satellite of the American National Aeronautics

and Space Administration (NASA). Like Envisat, Aura orbits the Earth in a near-polar sun-synchronous orbit; however, in contrast to MIPAS, it crosses the Equator at 13:45 local solar time on the ascending part of the orbit (Schoeberl et al., 2006). The operational phase of MLS started in August 2004 and is still ongoing. The instrument measures the thermal emission during day and night, using microwave radiometers operating at frequencies near 118, 190, 240, and 640 GHz, as well as a 2.5



**Table 1.** Measurement modes and data versions of MIPAS V8 ClO.

| Mode | Version | Height range / km |
|---|---|---|
| NOM, FR | V8_ClO_62 | 6–68 |
| NOM, RR | V8_ClO_262 | 7–72 |
| UTLS-1, RR | V8_ClO_162 | 5.5–49 |
| MA, RR | V8_ClO_562 | 18–102 |

THz module (Waters et al., 2006). ClO emissions are obtained from a line centered at 649.5 GHz. MLS provides a variety of vertical stratospheric temperature and composition profiles with a sampling rate of 3500 scans per day, which is nearly three times higher than the rate of MIPAS during the RR period.

The Earth system model EMAC uses the second version of the Modular Earth Submodel System (MESSy2) to link multi-
institutional computer codes. The core atmospheric model of EMAC is the fifth-generation European Centre Hamburg general circulation model (ECHAM5) (Röckner et al., 2006). For our study two EMAC simulations (MESSy version 2.55) were performed for 1–2 January and 1–2 July 2005, with a horizontal resolution of T42 (∼2.8° latitude × 2.8° longitude) on 90 vertical hybrid pressure levels from the surface up to 0.01 hPa. Each of these two simulations was initialized with corresponding rerun files from a longer EMAC simulation (1979 to 2022), which was nudged towards the ERA-5 reanalysis (Hersbach et al.,
2020) of the European Centre for Medium-range Weather Forecasts (ECMWF). A comprehensive chemistry setup was used.

## 3 Lessons learned from previous analyses

Generally, sub-millimeter and microwave measurements are the method of choice for remote measurements of ClO. Infrared measurements are much less sensitive, and most of the ClO transitions are largely overlapped by the signal of stronger infrared emitters/absorbers like $O_3$, $H_2O$, CFC-11 and $HNO_3$. Thus, retrieval of ClO volume mixing ratios in this wavenumber range
is a challenge.

First steps towards detection of ClO by Fourier Transform Infrared Spectroscopy were undertaken by Rinsland and Goldman (1992), who searched for suitable lines in the infrared region and cleared the way for MIPAS ClO evalution. Nevertheless, when the MIPAS mission was proposed, ClO was not mentioned on the list of detectable species, although its relevance to ozone destruction was well known (Fischer et al., 1988). However, Echle et al. (1992) indicated a chance for detection of
ClO by MIPAS under disturbed conditions. Indeed, Glatthor et al. (2004) subsequently showed that in the case of strong chlorine activation (ClO VMRs ≥ 2 ppbv) in the lower polar stratosphere, even single-profile ClO retrievals can successfully be performed (noise error 20%, total error 30%). These retrievals were confined to MIPAS FR measurements. ClO retrievals from MIPAS RR measurements based on version 4 and refined version 5 calibrated spectra were presented in von Clarmann et al. (2009b) and in von Clarmann et al. (2013), respectively. However, intercomparison of MIPAS ClO with data from
other satellite experiments in the framework of the Stratosphere-troposphere Processes And their Role in Climate (SPARC) data initiative (Hegglin and Tegtmeier, 2017) showed that the RR measurements were strongly biased high in the region of





the upper stratospheric VMR maximum at 35–40 km. Therefore MIPAS RR measurements at altitudes above 30 hPa were not presented in the final SPARC Data Initiative report. For MIPAS FR measurements this high bias was less distinct, but especially in the tropics it also amounted to more than 20%. Thus, one major objective for a MIPAS ClO update was to improve the upper stratospheric data quality. In addition, in this height range the vertical resolution of MIPAS ClO was quite poor. For this

reason, the spectral region for ClO retrieval in the upper stratosphere has been considerably enhanced in the V8 data version (see Section 5). For several years after launch, MIPAS data analysis by the European Space Agency (ESA) was restricted to the so-called main target species, but eventually an official ESA product of ClO also became available (Raspollini et al., 2022).

## 4    The mid-infrared signature of ClO

There is only one rather weak absorption band of chlorine monoxide in the mid-infrared region, namely the 0–1 fundamental at

844 cm$^{-1}$. Figure 1a,b illustrates the ClO signature in this spectral region, resulting from model calculations with the Karlsruhe Optimized and Precise Radiative Transfer Algorithm (KOPRA, Stiller 2000; Stiller et al. 2002) for a tangent altitude of 20 km and daytime southern polar winter conditions without and with inclusion of ClO. The modelled ClO VMR at the tangent altitude is 2.5 ppbv, which corresponds to very strong chlorine activation. Even for this scenario the strongest ClO lines are just of the order of the measurement noise in terms of noise equivalent spectral radiance (NESR) of 10 nW cm$^{-2}$ sr$^{-1}$ cm (Fig. 1b).

However, since more than 20 lines, i.e. nearly all lines shown, are used in the lower stratospheric MIPAS ClO retrieval, the signal-to-noise ratio is enhanced by at least the factor $\sqrt{20}$ to a value of ∼3.4 (assuming an average radiance of 7.5 nW cm$^{-2}$ sr$^{-1}$ cm of each of the 20 lines applied), which enables a reliable single-geolocation retrieval product. Figure 1c shows a zoom into the spectral region 831–837 cm$^{-1}$ with the radiance contribution of all gases, of ClO and of the interfering species CFC-11, O$_3$, HNO$_3$, CO$_2$, OCS and NO$_2$. This spectral region contains four prominent ClO lines at 832.0, 833.3, 834.63 and

835.95 cm$^{-1}$ - the latter three widely isolated and especially well suited for ClO retrieval.

The spectral signature of ClO at the altitude of 38 km, i.e. the region of the upper stratospheric ClO maximum, for mid-latitude summer day conditions and a ClO VMR of 0.55 ppbv is presented in Fig. 1d,e. At this altitude the radiance of the ClO lines is more than one order of magnitude lower than measurement noise. Although nearly all available ClO lines of the P-, Q-, and R-branches are used for retrieval, the signal-to-noise ratio is much lower, namely of the order of 0.3, taking into account

∼36 lines with an individual radiance of 0.5 nW cm$^{-2}$. Thus, averaging over dozens of ClO measurements is necessary to obtain reliable ClO VMRs in the region of the upper stratospheric maximum.

## 5    The retrieval of ClO

The underlying method of MIPAS trace gas retrievals is a constrained non-linear least squares fitting (von Clarmann et al., 2003, 2009b). The cost function that is to be minimized consists of two additive terms. The first one is the sum of squared

differences between the measured and the simulated spectrum, weighted by the inverse measurement error covariance matrix. The other term is a Tikhonov-type smoothness operator (Tikhonov, 1963), whose contribution to the cost function increases as





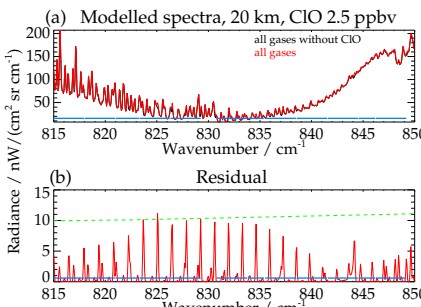 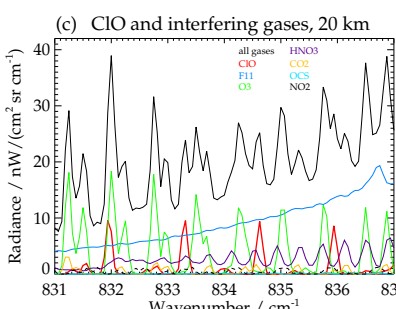 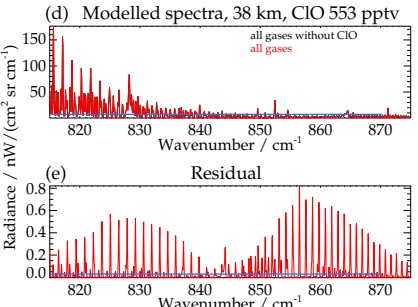

**Figure 1. (a)** KOPRA model spectra for southern polar winter day conditions and a tangent altitude of 20 km. The black curve, representing the radiance of all interfering gases without ClO, is mostly covered by the red curve, which illustrates the radiance including strongly enhanced ClO (2.5 ppbv at 20 km). Blue solid line: spectral range used for ClO retrieval; green dashed line: measurement noise in terms of noise equivalent spectral radiance (NESR). **(b)** Difference spectrum representing the ClO signature. **(c)** Zoom into the spectral region 831–837 $cm^{-1}$, showing the radiance contribution of all gases (black), of ClO (red) and of the interfering gases CFC-11 (blue), $O_3$ (green), $HNO_3$ (violet), $CO_2$ (orange), OCS (light blue), and $NO_2$ (dashed black). **(d)** Model spectra without (black) and with (red) ClO for northern mid-latitude summer day conditions, a tangent altitude of 38 km and a ClO VMR of 0.55 ppbv. **(e)** Difference spectrum representing the ClO signature.

the mixing ratio differences between adjacent gridpoints become more pronounced. Technical details of the implementation of the altitude-dependence of this side condition are discussed in Kiefer et al. (2021) and Kiefer et al. (2023). For the radiative transfer calculations and the calculation of the Jacobians needed, the radiative transfer model KOPRA is used.

The entire MIPAS retrieval is not performed in one run, but is decomposed species-wise, where, roughly speaking, each
species is analyzed in those spectral windows that contain the most information about that species and are the least affected by signals from interfering gases. This leads to a sequential retrieval where the contributors to the infrared spectrum which have the most prominent lines are retrieved first, and whose resulting volume mixing ratios are then treated as known parameters in subsequent retrieval steps. Accordingly, the MIPAS ClO retrieval uses V8 data from preceding retrievals of temperature and line-of-sight pointing, $ClONO_2$, CFC-12, $H_2O$, $HNO_3$, and $NO_2$. Version 8 ozone, which is also available, is used as first
guess and a priori of a combined ClO and $O_3$ retrieval. This is because $O_3$ interferences in the ClO analysis windows are so large that even minor spectroscopic inconsistencies between the lines used for the original ozone retrieval and those used in the ClO windows could have a sizable effect on the ClO results. Further, potential weak effects of non-local thermodynamic equilibrium (non-LTE) in these $O_3$ lines are also caught by this approach. The $O_3$ results from the combined ClO-$O_3$ retrieval are discarded because they are deemed inferior to the standard V8 $O_3$ results.
Compared to previous retrieval approaches, the uncertainties due to the VMRs of interfering species in a later position in the retrieval chain are reduced for most of the interferents, because in V8 ClO retrievals of NOM and UTLS-1 mode measurements (see Table 1) we use the actual V5 data of the relevant geolocation instead of climatological data (Kiefer et al., 2002), which were applied in earlier data versions. Version 5 volume mixing ratios are available for OCS, HCN, $C_2H_2$, $C_2H_6$, CFC-11, HCFC-22, CFC-113 and $HNO_4$. The origin of the prefits used in V8 ClO retrievals applying NOM and UTLS-1 mode spectra
is summarized in Fig. 2. The only exception to this procedure are the MA retrievals, because for these measurements V5 profiles



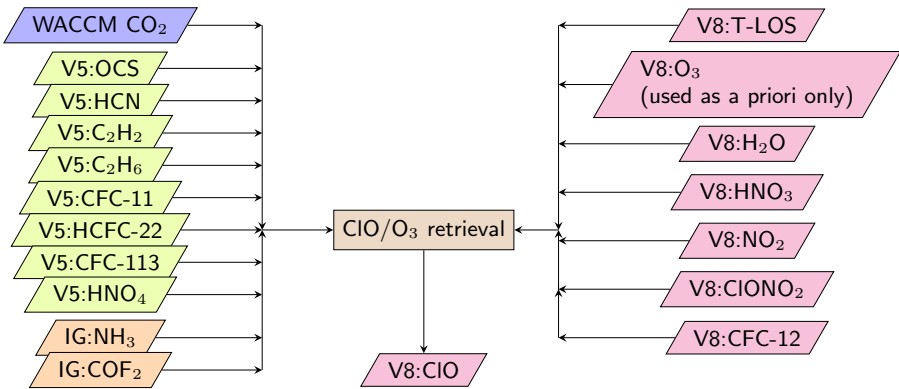

**Figure 2.** Data flow of the ClO retrieval. $CO_2$ profiles are generated using WACCM model output (Marsh, 2011; Marsh et al., 2013; García et al., 2017) as described by Kiefer et al. (2021). The origin of the other trace gases used for spectral modeling is colour coded. Green: V5 data, pink: V8 data, and light brown: climatology database (Kiefer et al., 2002).

are available for $H_2O$, $O_3$, $HNO_3$, and $NO_2$ only. For the other interferents required for evaluation of MA ClO, climatological mean data built from V5 NOM and UTLS-1 retrievals are applied.

Besides the V8 temperature profiles and gradients, three-dimensional temperature fields are used for the forward calculation as outlined in Kiefer et al. (2021). Information on the horizontal temperature structure is taken from ECMWF ERA-interim
analyses up to 60 km altitude and from NRLMSISE-00 (Picone et al., 2002) above. The horizontal temperature distributions are scaled to match the retrieved vertical temperature profile.

The microwindows chosen for the retrieval of ClO contain lines of the fundamental (0-1) band centered at 844 $cm^{-1}$ in MIPAS band A (685-980 $cm^{-1}$). In order to get more information and a better signal-to-noise ratio, MIPAS V8 retrievals of ClO use more microwindows (in total 17) than previous versions (Table 2). Up to 30 km they nearly completely cover the
P-, Q- and a part of the R-branch (815.0–849.1875 $cm^{-1}$, RR mode) and in addition the whole R-branch at higher altitudes (815.0–870.1875 $cm^{-1}$, RR mode). To maintain more flexibility in fitting wavenumber-dependent continuum-like features, these microwindows are not merged into one large analysis window, although they mostly are directly adjacent. While the P- and Q-branch lines of ClO are often disturbed by stronger $O_3$, $CO_2$ and $H_2O$ lines, the R-branch lines are even more severely disturbed by the broadband signature of CFC-11 and by strong $HNO_3$ lines. Therefore they are only used in the upper
stratosphere. At 12 km and below, the spectral regions 824.0625 – 830.75 $cm^{-1}$ and 839.0625 – 842.9375 $cm^{-1}$ (RR mode) are completely excluded, because they contain strong $H_2O$ lines at 825.1, 827.7, 839.9 and 841.9 $cm^{-1}$. Already in the V5 retrievals, where an automatically generated set of microwindows following the method by von Clarmann and Echle (1998) and Echle et al. (2000) was applied, these spectral regions were largely excluded at these altitudes. Further, the spectral range 829.3125 to 830.75 $cm^{-1}$ (RR mode) is completely discarded up to 30 km because of a weak Q-branch of $HNO_3$, which is not





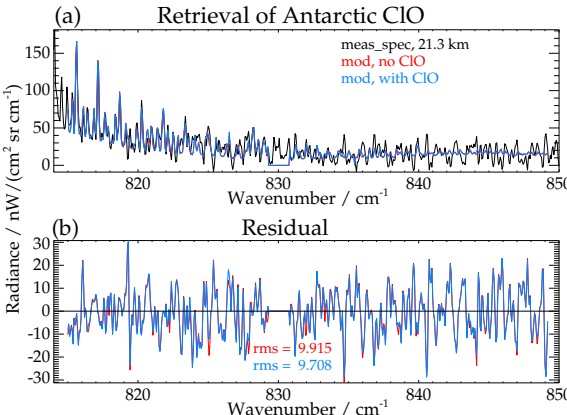

**Figure 3. (a)** Measured (black) and modelled (blue) spectra for a MIPAS observation of strongly enhanced ClO on 4 September 2007 in the Antarctic lower stratosphere (73.1°S, 162.1°W, 21.3 km). The second model spectrum (red), which is mostly covered by the blue one, results from an additional retrieval of all fit parameters used in the ClO retrieval except ClO itself. **(b)** Residuals between the model spectra and the measurement for retrieval with (blue) and without (red) inclusion of ClO. The respective RMS is indicated in the figure.

**Table 2.** Microwindows of the MIPAS V8 ClO retrieval

| Wavenumber range FR retrievals / cm$^{-1}$ | Wavenumver range RR retrievals / cm$^{-1}$ | Altitude range / km | Newly added |
|---|---|---|---|
| 815.000–817.950 | 815.0000–817.9375 | 6–68 | yes |
| 818.000–820.950 | 818.0000–820.9375 | 6–68 | yes |
| 821.000–823.275 | 821.0000–823.2500 | 6–68 | no |
| 823.300–826.275 | 823.3125–826.2500 | 6–68 | no |
| 826.300–830.775 | 826.3125–830.7500 | 15–68 | no |
| 830.800–835.400 | 830.8125–835.3750 | 6–68 | no |
| 835.500–838.525 | 835.5000–838.5000 | 6–68 | no |
| 838.550–841.525 | 838.5625–841.5000 | 6–68 | no |
| 841.550–846.200 | 841.5625–846.1875 | 6–68 | no |
| 846.250–849.175 | 846.2500–849.1875 | 6–68 | yes |
| 849.200–852.175 | 849.2500–852.1875 | 33–68 | yes |
| 852.200–855.175 | 852.2500–855.1875 | 33–68 | yes |
| 855.200–858.175 | 855.2500–858.1875 | 33–68 | yes |
| 858.200–861.175 | 858.2500–861.1875 | 33–68 | yes |
| 861.200–864.175 | 861.2500–864.1875 | 33–68 | yes |
| 864.200–867.175 | 864.2500–867.1875 | 33–68 | yes |
| 867.200–870.175 | 867.2500–870.1875 | 33–68 | yes |

modelled properly using the HNO$_3$ profiles prefitted at higher wavenumbers. The microwindows used for FR-mode retrievals are essentially the same, with their boundaries shifted to the nearest FR spectral gridpoint, where necessary.





While for MIPAS V5 retrievals spectroscopic ClO data from the HITRAN-1996 database (Rothman et al., 1998) were used, for V8 ClO retrievals the HITRAN-2016 spectroscopy (Gordon et al., 2017) is applied. ClO line intensites in the latter data base are about 10% weaker than those used before, leading to an increase of the retrieved VMRs of the same order.

Besides the VMRs of ClO and ozone, a background continuum and an offset correction, as suggested by von Clarmann et al.

(2003), are also fitted to the spectra. In older MIPAS data versions, the background continuum was set to zero above 32 km via a hard constraint, because it was believed that above the Junge layer (Junge and Manson, 1961) no appreciable amounts of particle matter existed that could cause any sizable continuum emission. A relevant continuum-like signal also cannot be explained by the far wings of the weak spectral lines prevailing at these altitudes. Nevertheless, it was found that allowing continuum emission from higher altitudes led to more plausible VMR profiles of many trace gases (see, e.g., Haenel et al.,

2015, or Kiefer et al., 2021). Thus, in V8 ClO retrievals the upper boundary of the background continuum is set to 58 km altitude. The improved offset retrieval scheme, consisting of microwindow-dependent constrained offset profiles, presented by Kiefer et al. (2021) was also applied to ClO.

The retrieved ClO profile is represented on a regular vertical grid independent of the tangent altitudes of the limb scan under assessment. The grid is the same for all observation modes, with a width of 1 km between 4 and 60 km. Between 60 and 70

km the width is 2 km, and additional levels are at the altitudes of 75, 80, 90, 100, and 120 km. Since the retrieval grid is finer than the tangent altitude grid of the measurements, the inverse problem is inherently ill-posed and needs regularization. We use a regularization involving a squared first-order finite differences matrix as formalized by, e.g., Tikhonov (1963), Twomey (1963) or Phillips (1962). This regularization method smooths the retrieval but does not push it towards a priori assumptions on the atmospheric ClO content. The strength of the smoothing constraint is chosen to be altitude-dependent, with stronger

regularization at altitudes where ClO VMRs are expected to be low. The former implementation of this altitude dependence has been replaced by that described by Kiefer et al. (2021, their Eq. 3). Above 20 km, the new regularization is weaker than that employed in previous versions, leading to better vertical resolution, but to somewhat more oscillatory profiles. The a priori VMR profile chosen for the ClO retrieval is a flat all-zero profile. In addition to the Tikhonov-type smoothing constraint, the regularization matrix contains diagonal terms at the two uppermost altitudes. The purpose of these terms is to avoid unphysical

values far below zero for the upper parts of the profiles. The averaging kernels (see Section 7.1) contain all the information needed to judge the impact of the regularization on the resulting profiles. The a priori profile for the joint fitted ozone is the standard V8 ozone profile.

## 6   Significance test

Figure 3a shows the measured (black) and the modelled (blue) spectrum for an observation of strongly enhanced ClO (2.65

ppbv) on 4 September 2007 at 21.3 km altitude in the Antarctic lower stratosphere (73.13°S, 162.09°W, solar elevation angle 6.67°). The additional red spectrum, which is mostly covered by the blue one, results from a second retrieval for a ClO-free atmosphere using all fit parameters except ClO itself. The retrieval without ClO leads to a worsening of the root-mean-square (RMS) deviation between the modelled and measured spectrum by 2% (from 9.71 to 9.92 nW cm$^{-2}$ sr$^{-1}$ cm). The missing





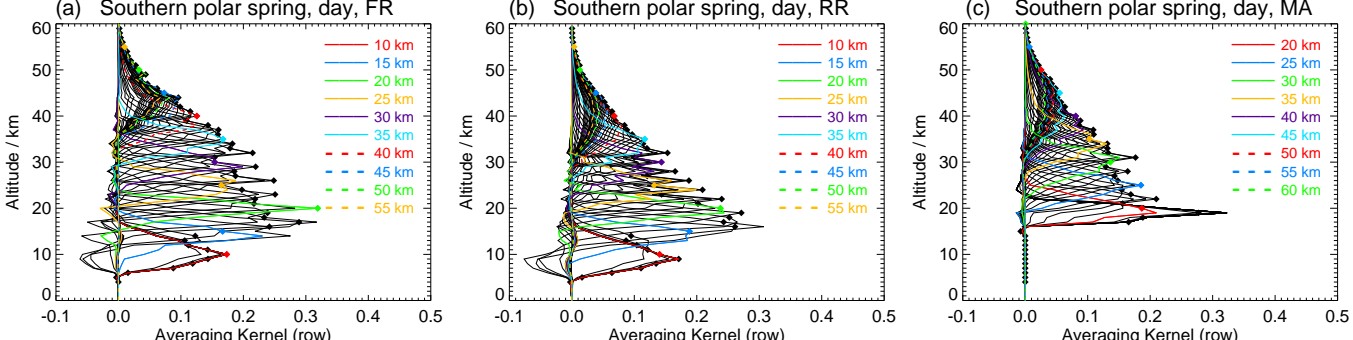

**Figure 4. (a)** Averaging kernels related to a V8 ClO retrieval from a FR measurement. The limb scan of this example was recorded at 58.28°S, 31.58°E on 2 September 2002 during Envisat orbit 2651 under a solar elevation angle of 19.08°. **(b)** Same as (a), but for a ClO RR NOM measurement at 62.48°S, 158.12°E on 4 September 2007 during Envisat orbit 28825 under a solar elevation angle of 16.14°. **(c)** Same as (a), but for a ClO RR MA measurement at 70.63°S, 111.58°E on 13 September 2007 during Envisat orbit 28943 under a solar elevation angle of 12.30°S.

ClO, which cannot be compensated by the remaining fit parameters, is visible in the red negative spikes in the residual spectrum (Fig. 3b). This result demonstrates the significance of single-geolocation ClO retrievals in the case of strong chlorine activation in the polar lower stratosphere.

## 7 Data characterization

### 7.1 Averaging kernels

We use averaging kernels to characterize the content of a priori information and the vertical resolution of the ClO retrievals, following the methodological framework by Rodgers (2000). Rows of the averaging kernel matrix are shown in Fig. 4 for FR, RR NOM, and RR MA measurements, with selected altitudes indicated by individual colours. Here, polar winter examples are chosen, because this is the situation where MIPAS ClO measurements are considered most relevant. At altitudes below 40 km the averaging kernels are fairly well-behaved in the sense that they peak at or close to the nominal altitudes, which are indicated by diamonds. This means that the regularization of the inversion causes little or no vertical information displacement. FR and RR averaging kernels are fairly similar. Extremely low values of the averaging kernel diagonals at about 50 km and above indicate that MIPAS does not contain appreciable information on ClO abundances at these altitudes.

### 7.2 Horizontal distribution of information

The asumption of local spherical homogeneity of the atmosphere affects the horizontal resolution of the retrieval. We use the concept of horizontal averaging kernels as suggested by von Clarmann et al. (2009a) to characterize related effects. In





**Table 3.** Horizontal information distribution of MIPAS ClO V8 data.

| Altitude / km | FR-smearing / km | FR-displacement / km | RR-smearing / km | RR-displacement / km | MA-smearing / km | MA-displacement / km |
|---|---|---|---|---|---|---|
| 66 | 465 | -78 | 452 | -29 | 431 | 52 |
| 60 | 465 | -78 | 452 | -29 | 431 | 52 |
| 55 | 466 | -77 | 452 | -29 | 431 | 52 |
| 50 | 468 | -76 | 454 | -28 | 435 | 53 |
| 45 | 475 | -70 | 463 | -25 | 449 | 58 |
| 40 | 482 | -56 | 486 | -20 | 449 | 68 |
| 35 | 404 | -24 | 465 | -13 | 430 | 77 |
| 30 | 374 | 17 | 403 | 6 | 381 | 89 |
| 25 | 353 | 57 | 369 | 29 | 356 | 105 |
| 20 | 339 | 93 | 327 | 58 | | |
| 15 | 364 | 129 | 351 | 92 | | |
| 10 | 397 | 152 | 367 | 108 | | |

Table 3 we show the horizontal information smearing and displacement for the ClO retrieval, determined by the use of this concept. The horizontal smearing is calculated as the full width at half maximum of the vertically integrated 2D averaging kernels. The horizontal information displacement is the distance between the averaging-kernel-weighted mean of the horizontal coordinate and the nominal geolocation of the limb scan. The sign convention is such that positive values indicate information displacements towards the satellite. For all measurement modes, the horizontal smearing is on the order of the size of the horizontal latitudinal sampling, i.e., the horizontal distance between the nominal geolocations of two subsequent limb scans. The displacement is less than 100 km except for the bottom ends of the profiles.

### 7.3 Volume mixing ratios, measurement noise and vertical resolution along selected orbits

Figure 5 shows the volume mixing ratio, the measurement noise and the vertical resolution of V8 ClO for MIPAS FR measurements on 31 August 2002 along orbit 2624 (left column), for RR NOM measurements on 4 September 2007 along orbit 28825 (middle column), and for RR MA measurements on 13 September 2008 along orbit 34191 (right column). The white areas at the bottom of the plots of the FR and RR NOM measurements are excluded due to filtering for clouds. Caused by polar stratospheric clouds, this area extends up to 20 km altitude in the Antarctic. Further, there are also high clouds in the equatorial region. Due to the scan pattern, the RR MA distributions are generally restricted to altitudes above 18 km. The upper parts of the profiles above 46–50 km are excluded, because the retrieval contains little information in this height region.

For each measurement mode, strong ClO enhancements of up to more than 2.5 ppbv, extending over 5–7 consecutive scans, are visible around 20 km altitude at the edge of the Antarctic vortex (Fig. 5a–c). These enhancements are caused by heterogeneous reactions of the reservoir species HCl and ClONO$_2$ on polar stratospheric clouds and photolysis of the released Cl$_2$ after the end of polar night (Solomon et al., 1986; Solomon, 1999). In all other regions of the lower and middle stratosphere (15-30 km), the ClO VMRs scatter around zero. In the altitude region of the upper stratospheric ClO maximum (35–40 km), the VMRs retrieved from daytime measurements are around 0.5 ppbv. The latter values exhibit large scatter between single-scan



**Figure 5.** ClO distributions from MIPAS FR measurements on 31 August 2002 along orbit 2624 (left column), from RR NOM measurements on 4 September 2007 along orbit 28825 (middle column), and from RR MA measurements on 13 September 2008 along orbit 34191 (right column). **(a–c)** ClO volume mixing ratio. **(d–f)** Estimated standard deviation (ESD). **(g–i)** Vertical resolution in terms of the full width at half maximum of the rows of the averaging kernel. Nighttime measurements are indicated by red crosses and daytime measurements by red plus signs at the bottom of the plots. White areas at the lower end of the latitudinal cross sections are data gaps due to clouds, in particular polar stratospheric clouds (PSCs) near the South Pole. MA measurements are generally restricted to altitudes above 18 km.





**Table 4.** The vertical resolution of MIPAS ClO V5 and V8 data in terms of the full width at half maximum of the rows of the averaging kernel.

| Altitude / km | V5 ClO, FR / km | V8 ClO, FR / km | V5 ClO, RR NOM / km | V8 ClO, RR NOM / km | V8 ClO, RR MA / km |
|---|---|---|---|---|---|
| 10 | 5 | 5 | 5 | 4.5 | |
| 18 | 4 | 4 | 4 | 4 | 3.5 |
| 20 | 4 | 4 | 4 | 4 | 4 |
| 30 | 7.5 | 5.5 | 7 | 5.5 | 6 |
| 40 | 14 | 7.5 | 17 | 9.5 | 9.5 |
| 50 | 12 | 10 | 12.5 | 12.5 | 13 |

measurements, but after averaging over longer time periods, e.g. monthly means, the upper stratospheric maximum is clearly visible (see Figs. 10, 11).

Figure 5d–f shows the measurement noise propagated into the ClO retrieval in terms of estimated standard deviation (ESD). Outside the region of the Antarctic vortex, the ESD increases from 0.3 ppbv around 20 km to 0.55 ppbv (FR mode) and 0.65 ppbv (RR modes) at 50 km. In the region of the Antarctic vortex, higher ESD values of 0.5–0.8 ppbv generally cover the entire altitude region from 20 to 50 km, especially for the RR modes. However, for the FR and the RR NOM mode, the ESD in the region of the lower stratospheric maxima, which are situated at the edge of the Antarctic vortex, is only around 0.3–0.5 ppbv.

The vertical resolution in terms of full-width at half maximum of the rows of the averaging kernels (Fig. 5g–i) is 4 km around 20 km and 5.5–6 km at 30 km altitude for both measurement modes. At higher altitudes the height resolution degrades, to 7.5 km at 40 km and to ∼10 km at 50 km for FR measurements. For the RR observations the decline is stronger, namely to 9.5 km at 40 km and to 12.5–13 km at 50 km.

A summary of the vertical resolution of MIPAS V5 and V8 data for the FR and RR NOM and MA measurement modes is given in Table 4. For MA measurements, V5 data are not available. The vertical resolution of V8 ClO has improved considerably at altitudes of 30 km and above due to the larger number of microwindows used for the retrieval. The improvement is largest at the altitude of the upper stratospheric ClO maximum around 40 km. At 50 km the difference in vertical resolution becomes smaller again due to the apparently better height resolution (compared to that at 40 km) of the V5 data. But at this altitude the V5 averaging kernels (not shown) are even more vertically displaced and asymmetric than the V8 kernels.

### 7.4 Error budget

The general approach to error estimation is described in detail by von Clarmann et al. (2022) and is compliant with the recommendations of the 'Towards Unified Error Reporting (TUNER)' activity (von Clarmann et al., 2020). The relevant error sources and their magnitudes are summarized in Tables 5, 6 and 7 for high-resolution NOM, reduced-resolution NOM and reduced-resolution MA measurements, respectively. All error estimates are given in terms of $1\sigma$.

The resulting error estimates of the ClO VMRs for southern polar winter day conditions are presented in Fig. 6 for FR measurements (left), RR-NOM measurements (middle) and RR-MA measurements (right). To present more representative values, averaging over the error estimates of about 30 geolocations was performed in all cases. At every altitude the total





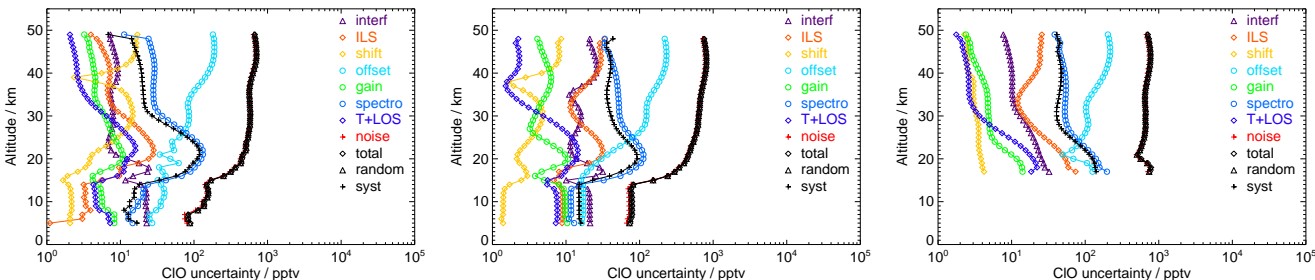

**Figure 6.** Left: Uncertainties of ClO mixing ratios retrieved from MIPAS FR measurements for southern polar winter day conditions. The lines and symbols representing the measurement noise, the random error and the total error are overlaid and thus hardly distinguishable. No error estimates are provided for altitudes above 50 km, because MIPAS is not sensitive to ClO there. Middle: Same as left, but for MIPAS RR measurements in NOminal Mode (NOM). Right: Same as left, but for MIPAS RR measurements in Middle Atmosphere mode (MA).

error is dominated by the random error, which is almost identical to measurement noise, with a considerable gap followed by uncertainties in spectroscopic parameters, in the instrumental offset and in the internal line shape (ILS). Concerning the last three error sources, at the altitude of 20 km the spectroscopic error is most relevant, but at the altitude of 40 km the offset uncertainty is dominating. The total error is lowest (60–150 pptv) in the troposphere, where, however, hardly any ClO

is present. In the height region of the polar lower stratospheric ClO maximum (20 km), the total error is 400-500 pptv, which makes ClO enhancements of 1.5 ppbv or more well detectable in single MIPAS observations, implying that even single-profile ClO VMRs can be used to study stratospheric chlorine chemistry and to identify regions of chlorine activation. In the region around 40 km, the total error is 700–800 pptv, which is about 200 pptv higher than the upper stratospheric ClO maximum. However due to the dominance of the noise error, the errors average down when, e.g., zonal mean concentrations, averages in

latitude-longitude bins, or similar statistical estimates are calculated. Error estimates for the other scenarios are shown in the Supplement.

### 7.5 Coarse grid representation

The regularization of the MIPAS standard retrievals implies that the VMRs on the vertical retrieval grid are not independent of each other. As a consequence, the data user will often have to utilize averaging kernels. For certain applications, such as

time series analysis, even the deployment of averaging kernels does not provide an obvious solution to the challenge that the vertical resolution can change from profile to profile. In order to avoid problems of this type, we offer an alternative data product where the related averaging kernels are unity; i.e. the profiles are free of formal a priori information, and the vertical resolution is defined entirely by the vertical grid chosen. This representation is meant to complement the standard data product rather than to replace it. Technically, a vertical grid is constructed that is coarse enough to allow stable retrievals without

regularization, following the method proposed by von Clarmann and Grabowski (2007). The coarse grid uses pressure as the vertical coordinate and is the same for all limb scans. The new gridpoints are 1000, 700, 500, 300, 170, 100, 50, 30, 15, 7, 3, 1, 0.1, 0.01, 0.001, and $3\times10^{-5}$ hPa, for both FR and RR retrievals. The transformation to this grid is performed as described



**Table 5.** Assumed uncertainties for northern mid-latitude summer daytime conditions affecting the retrieval of ClO (FR).

| Type of uncertainty | Value / Typical value[1] | Source[2] | Propagation Method[3] |
|---|---|---|---|
| noise | 30 nW/(cm$^2$ sr cm$^{-1}$) | Kleinert et al. (2018) | G(16) |
| offset | 6 nW/ (cm$^2$ sr cm$^{-1}$) | Kleinert et al. (2018) | G(5;13) |
| gain random | 0.21% | Kleinert et al. (2018) | P(21;22) |
| gain systematic | 1.15% | Kleinert et al. (2018) | P(21;22) |
| spectral shift | 0.00029 cm$^{-1}$ | preceding V8 retrieval (Kiefer et al., 2021) | P(7) |
| ILS | 3% | Hase (2003) | P(7;14;15) |
| Temperature, noise | 0.24 - 1.04 K | preceding V8 retrieval (Kiefer et al., 2021) | G(6) |
| Tangent altitudes, noise | 35 - 79 m | preceding V8 retrieval (Kiefer et al., 2021) | G(6) |
| Temperature, spectral shift | 0.01 - 0.71 K | preceding V8 retrieval (Kiefer et al., 2021) | P(17) |
| Tangent altitudes, spectral shift | 2 - 41 m | preceding V8 retrieval (Kiefer et al., 2021) | P(17) |
| Temperature, offset | 0.05 - 0.39 K | preceding V8 retrieval (Kiefer et al., 2021) | P(7) |
| Tangent altitudes, offset | 8 - 24 m | preceding V8 retrieval (Kiefer et al., 2021) | P(7) |
| Temperature, ILS | 0.03 - 1.26 K | preceding V8 retrieval (Kiefer et al., 2021) | P(7) |
| Tangent altitudes, ILS | 7 - 122 m | preceding V8 retrieval (Kiefer et al., 2021) | P(7) |
| Temperature, $CO_2$ intensities | 0.03 - 0.16 K | preceding V8 retrieval (Kiefer et al., 2021) | P(7;26;27) |
| Tang. alt., $CO_2$ intensities | 26 - 34 m | preceding V8 retrieval (Kiefer et al., 2021) | P(7;26;27) |
| Temperature, $CO_2$ broadening | 0.09 - 1.10 K | preceding V8 retrieval (Kiefer et al., 2021) | P(7;28;29) |
| Tang. alt., $CO_2$ broadening | 198 - 252 m | preceding V8 retrieval (Kiefer et al., 2021) | P(7;28;29) |
| Temperature, gain, syst. | 0.30 - 0.79 K | preceding V8 retrieval (Kiefer et al., 2021) | P(21) |
| Tangent altitudes, gain, syst. | 2 - 51 m | preceding V8 retrieval (Kiefer et al., 2021) | P(21) |
| Temperature, gain, random | 0.05 - 0.15 K | preceding V8 retrieval (Kiefer et al., 2021) | P(21) |
| Tangent altitudes, gain, random | 0.4 - 9 m | preceding V8 retrieval (Kiefer et al., 2021) | P(21) |
| vmr(CFC-12) | 8.29E-06 - 1.14E-04 ppmv | preceding V8 retrieval (Stiller et al., 2024) | G(6) |
| vmr($ClONO_2$) | 9.85E-06 - 8.20E-05 ppmv | preceding V8 retrieval (Stiller et al., 2025) | G(6) |
| vmr($H_2O$) | 1.60E-01 - 2.54E+00 ppmv | preceding V8 retrieval (Kiefer et al., 2024) | G(6) |
| vmr($HNO_3$) | 6.18E-05 - 4.78E-04 ppmv | preceding V8 retrieval (Stiller et al., 2025) | G(6) |
| vmr($NO_2$) | 3.09E-07 - 2.27E-04 ppmv | preceding V8 retrieval (Funke et al., 2024) | G(6) |
| vmr(CFC-11) | 2.68E-08 - 5.77E-06 ppmv | V5 retrieval (Kellmann et al., 2012) | G(6) |
| vmr(CFC-113) | 2.17E-07 - 1.21E-05 ppmv | V5 retrieval (unpublished data) | G(6) |
| vmr($COF_2$) | 7.02E-07 - 3.70e-05 ppmv | V5 retrieval unpublished data | P(7;11) |
| vmr($C_2H_2$) | 2.98E-06 - 1.01E-05 ppmv | V5 retrieval (Glatthor et al., 2007) | G(6) |
| vmr($C_2H_6$) | 2.85E-07 - 8.69E-05 ppmv | V5 retrieval (Glatthor et al., 2007) | G(6) |
| vmr(HCFC-22) | 4.51E-06 - 1.43E-05 ppmv | V5 retrieval (Chirkov et al., 2016) | G(6) |
| vmr(HCN) | 1.50E-05 - 4.40E-05 ppmv | V5 retrieval (Glatthor et al., 2015) | G(6) |
| vmr($HNO_4$) | 9.81E-06 - 1.66E-04 ppmv | V5 retrieval (Stiller et al., 2007) | G(6) |
| vmr(OCS) | 3.85E-05 - 2.11E-04 ppmv | V5 retrieval (Glatthor et al., 2017) | G(6) |
| vmr($CO_2$) | 7.56E-01 - 7.51E-00 ppmv | WACCM model calculation (Kiefer et al., 2021) | P(20) |
| vmr($NH_3$) | 5.74E-11 - 3.00e-04 ppmv | climatology database (Kiefer et al., 2002) | P(7;11) |
| spectroscopic data (intensities) | 7.5% | HITRAN (Gordon et al., 2017) | P(7;11) |
| spectroscopic data (widths) | 7.5% | HITRAN (Gordon et al., 2017) | P(7;11) |

[1] For variable uncertainties typical values are reported. Some errors given in Kleinert et al. (2018) in terms of $2\sigma$ have been rescaled to $1\sigma$. [2] In cases where V5 data are unpublished, reference to publications of an earlier data version is made. [3] Temperature and tangent altitude errors due to noise, spectral shift, offset, instrumental line shape (ILS), gain calibration and $CO_2$ spectroscopy are propagated separately. The notation is as follows: "G" refers to generalized Gaussian error propagation in a matrix formalism, while "P" refers to perturbation studies. The numbers in parentheses are the respective equation numbers in von Clarmann et al. (2022).





**Table 6.** Assumed uncertainties for northern mid-latitude summer daytime conditions affecting the retrieval of ClO (RR NOM). See footnotes of Table 5 for details on the contents of the columns.

| Type of uncertainty | Value / Typical value | Source | Propagation Method |
|---|---|---|---|
| noise | 20 nW/(cm$^2$ sr cm$^{-1}$) | Kleinert et al. (2018) | G(16) |
| offset | 3 nW/ (cm$^2$ sr cm$^{-1}$) | Kleinert et al. (2018) | G(5;13) |
| gain random | 0.21% | Kleinert et al. (2018) | P(21;22) |
| gain systematic | 1.15% | Kleinert et al. (2018) | P(21;22) |
| spectral shift | 0.00029 cm$^{-1}$ | preceding V8 rieval (Kiefer et al., 2021) | P(7) |
| ILS | 3% | Hase (2003) | P(7;14;15) |
| Temperature, noise | 0.22 - 1.23 K | preceding V8 retrieval (Kiefer et al., 2021) | G(6) |
| Tangent altitudes, noise | 29 - 52 m | preceding V8 retrieval (Kiefer et al., 2022) | G(6) |
| Temperature spectral shift | 0.01 - 0.10 K | preceding V8 retrieval (Kiefer et al., 2021) | P(17) |
| Tangent altitudes, spectral shift | 1 - 15 m | preceding V8 retrieval (Kiefer et al., 2022) | P(17) |
| Temperature, offset | 0.03 - 0.45 K | preceding V8 retrieval (Kiefer et al., 2021) | P(7) |
| Tangent altitudes, offset | 8 - 16 m | preceding V8 retrieval (Kiefer et al., 2022) | P(7) |
| Temperature, ILS | 0.05 - 1.16 K | preceding V8 retrieval (Kiefer et al., 2021) | P(7) |
| Tangent altitudes, ILS | 8 - 113 m | preceding V8 retrieval (Kiefer et al., 2021) | P(7) |
| Temperature, ILS | 0.05 - 1.16 K | preceding V8 retrieval (Kiefer et al., 2021) | P(7) |
| Tangent altitudes, ILS | 8 - 113 m | preceding V8 retrieval (Kiefer et al., 2021) | P(7) |
| Temperature, $CO_2$ intensities | 0.03 - 0.18 K | preceding V8 retrieval (Kiefer et al., 2021) | P(7;26;27) |
| Tang. alt., $CO_2$ intensities | 26 - 34 m | preceding V8 retrieval (Kiefer et al., 2021) | P(7;26;27) |
| Temperature, $CO_2$ broadening | 0.14 - 1.47 K | preceding V8 retrieval (Kiefer et al., 2021) | P(7;28;29) |
| Tang. alt., $CO_2$ broadening | 198 - 252 m | preceding V8 retrieval (Kiefer et al., 2021) | P(7;28;29) |
| Temperature, gain, syst. | 0.2 - 0.8 K | preceding V8 retrieval (Kiefer et al., 2021) | P(21) |
| Tangent altitudes, gain, syst. | 1 - 49 m | preceding V8 retrieval (Kiefer et al., 2021) | P(21) |
| Temperature, gain, random | 0.04 - 0.15 K | preceding V8 retrieval (Kiefer et al., 2021) | P(21) |
| Tangent altitudes, gain, random | 0.2 - 9 m | preceding V8 retrieval (Kiefer et al., 2021) | P(21) |
| vmr(CFC-12) | 1.13E-05 - 1.19E-04 ppmv | preceding V8 retrieval (Stiller et al., 2024) | G(6) |
| vmr($ClONO_2$) | 9.99E-06 - 8.28E-05 ppmv | preceding V8 retrieval (Stiller et al., 2025) | G(6) |
| vmr($H_2O$) | 1.87E-01 - 2.24E+00 ppmv | preceding V8 retrieval (Kiefer et al., 2024) | G(6) |
| vmr($HNO_3$) | 5.54E-05 - 4.51E-04 ppmv | preceding V8 retrieval (Stiller et al., 2025) | G(6) |
| vmr($NO_2$) | 2.03E-07 - 2.88E-04 ppmv | preceding V8 retrieval (Funke et al., 2024) | G(6) |
| vmr(CFC-11) | 2.64E-08 - 5.94E-06 ppmv | V5 retrieval (Kellmann et al., 2012) | G(6) |
| vmr(CFC-113) | 2.26E-07 - 1.14E-05 ppmv | V5 retrieval (unpublished data) | G(6) |
| vmr($C_2H_2$) | 3.22E-06 - 9.55E-06 ppmv | V5 retrieval (Glatthor et al., 2007) | G(6) |
| vmr($C_2H_6$) | 2.77E-07 - 7.56E-05 ppmv | V5 retrieval (Glatthor et al., 2007) | G(6) |
| vmr(HCFC-22) | 4.78E-06 - 1.46E-05 ppmv | V5 retrieval (Chirkov et al., 2016) | G(6) |
| vmr(HCN) | 1.54E-05 - 4.41E-05 ppmv | V5 retrieval (Glatthor et al., 2015) | G(6) |
| vmr($HNO_4$) | 1.42E-05 - 1.66E-04 ppmv | V5 retrieval (von Clarmann et al., 2013) | G(6) |
| vmr(OCS) | 3.37E-05 - 2.14E-04 ppmv | V5 retrieval (Glatthor et al., 2017) | G(6) |
| vmr($CO_2$) | 7.56E-01 - 7.51E-00 ppmv | WACCM model calculation (Kiefer et al., 2021) | P(20) |
| vmr($COF_2$) | 6.67E-06 - 6.94e-05 ppmv | climatology database (Kiefer et al., 2002) | P(7;11) |
| vmr($NH_3$) | 5.74E-11 - 3.00e-04 ppmv | climatology database (Kiefer et al., 2002) | P(7;11) |
| spectroscopic data (intensities) | 7.5% | HITRAN (Gordon et al., 2017) | P(7;11) |
| spectroscopic data (widths) | 7.5% | HITRAN (Gordon et al., 2017) | P(7;11) |



**Table 7.** Assumed uncertainties for northern mid-latitude summer daytime conditions affecting the retrieval of ClO (RR MA). See footnotes of Table 5 for details on the contents of the columns.

| Type of uncertainty | Value / Typical value | Source | Propagation Method |
|---|---|---|---|
| noise | 20 nW/(cm$^2$ sr cm$^{-1}$) | Kleinert et al. (2018) | G(16) |
| offset | 3 nW/ (cm$^2$ sr cm$^{-1}$) | Kleinert et al. (2018) | G(5;13) |
| gain random | 0.21% | Kleinert et al. (2018) | P(21;22) |
| gain systematic | 1.15% | Kleinert et al. (2018) | P(21;22) |
| spectral shift | 0.00029 cm$^{-1}$ | preceding V8 retrieval (Kiefer et al., 2021) | P(7) |
| ILS | 3% | Hase (2003) | P(7;14;15) |
| Temperature, noise | 0.22 - 2.24 K | preceding V8 retrieval (García-Comas et al., 2023) | G(6) |
| Tangent altitudes, noise | 27 - 47 m | preceding V8 retrieval (García-Comas et al., 2023) | G(6) |
| Temperature, spectral shift | 0.004 - 0.11 K | preceding V8 retrieval (García-Comas et al., 2023) | P(17) |
| Tang. alt., spectral shift | 0.4 - 14 m | preceding V8 retrieval (García-Comas et al., 2023) | P(17) |
| Temperature, offset | 0.05 - 0.72 K | preceding V8 retrieval (García-Comas et al., 2023) | P(7) |
| Tangent altitudes, offset | 1 - 16 m | preceding V8 retrieval (García-Comas et al., 2023) | P(7) |
| Temperature, ILS | 0.08 - 0.78 K | preceding V8 retrieval (García-Comas et al., 2023) | P(7) |
| Tangent altitudes, ILS | 0.5 - 83 m | preceding V8 retrieval (García-Comas et al., 2023) | P(7) |
| Temp., CO2 intensities | 0.02 - 0.12 K | preceding V8 retrieval (García-Comas et al., 2023) | P(7;26;27) |
| Tang. alt., CO2 intensities | 20 - 35 m | preceding V8 retrieval (García-Comas et al., 2023) | P(7;26;27) |
| Temp., CO2 broadening | 0.08 - 1.17 K | preceding V8 retrieval (García-Comas et al., 2023) | P(7;28;29) |
| Tang. alt., CO2 broadening | 14 - 242 m | preceding V8 retrieval (García-Comas et al., 2023) | P(7;28;29) |
| Temperature, gain, syst. | 0.26 - 0.48 K | preceding V8 retrieval (García-Comas et al., 2023) | P(21) |
| Tangent altitudes, gain, syst. | 0.3 - 32 m | preceding V8 retrieval (García-Comas et al., 2023) | P(21) |
| Temperature, gain, random | 0.05 - 0.09 K | preceding V8 retrieval (García-Comas et al., 2023) | P(21) |
| Tang. alt., gain, random | 0.06 - 6 m | preceding V8 retrieval (García-Comas et al., 2023) | P(21) |
| vmr(CFC-12) | 9.64E-06 - 1.22E-04 ppmv | preceding V8 retrieval (Stiller et al., 2024) | G(6) |
| vmr(ClONO$_2$) | 1.12E-05 - 8.03E-05 ppmv | preceding V8 retrieval (Stiller et al., 2025) | P(7;11) |
| vmr(H$_2$O) | 1.45E-01 - 7.31E-01 ppmv | preceding V8 retrieval (Kiefer et al., 2024) | G(6) |
| vmr(HNO$_3$) | 5.99E-05 - 4.23E-04 ppmv | preceding V8 retrieval (Stiller et al., 2025) | G(6) |
| vmr(NO$_2$) | 1.19E-05 - 2.29E-04 ppmv | preceding V8 retrieval (Funke et al., 2024) | G(6) |
| vmr(CO$_2$) | 7.56E-01 - 7.51E+00 ppmv | WACCM model calc. (García-Comas et al., 2023) | P(20) |
| vmr(CFC-11) | 9.68E-12 - 2.57E-05 ppmv | climatology database (Kiefer et al., 2002)[1] | P(7;11) |
| vmr(CFC-113) | 1.26E-10 - 7.79E-05 ppmv | climatology database (Kiefer et al., 2002) | P(7;11) |
| vmr(HCFC-22) | 3.37E-07 - 7.45E-06 ppmv | climatology database (Kiefer et al., 2002) | P(7;11) |
| vmr(C$_2$H$_2$) | 0.00E+00 - 8.34E-08 ppmv | climatology database (Kiefer et al., 2002) | P(7;11) |
| vmr(C$_2$H$_6$) | 2.83E-15 - 5.92E-05 ppmv | climatology database (Kiefer et al., 2002) | P(7;11) |
| vmr(COF$_2$) | 9.35E-06 - 6.94E-05 ppmv | climatology database (Kiefer et al., 2002) | P(7;11) |
| vmr(HCN) | 1.70E-05 - 2.42E-04 ppmv | climatology database (Kiefer et al., 2002) | P(7;11) |
| vmr(HNO$_4$) | 6.30E-08 - 1.59E-04 ppmv | climatology database (Kiefer et al., 2002) | P(7;11) |
| vmr(NH$_3$) | 5.74E-11 - 1.25E-04 ppmv | climatology database (Kiefer et al., 2002) | P(7;11) |
| vmr(OCS) | 7.29E-09 - 3.09E-04 ppmv | climatology database (Kiefer et al., 2002) | P(7;11) |
| spectrosc. data (intensities) | 7.5% | HITRAN (Gordon et al., 2017) | P(7;11) |
| spectrosc. data (widths) | 7.5% | HITRAN (Gordon et al., 2017) | P(7;11) |

[1] Since V5 data are not available for MA measurements, for CFC-11 and the subsequent gases climatological uncertainties were used (Kiefer et al., 2002).





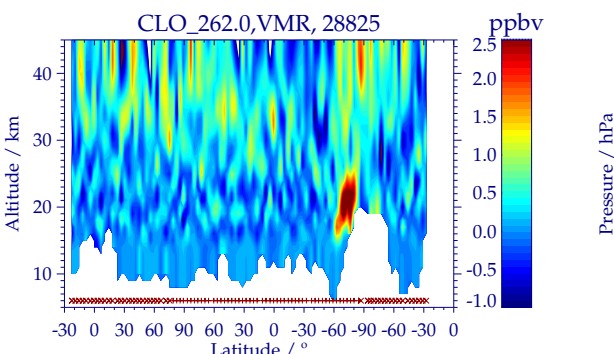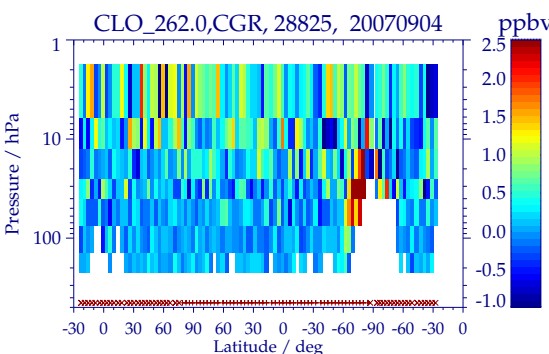

**Figure 7.** Regular (left) and coarse grid (right) representation of MIPAS V8 ClO, measured on 4 September 2007 during Envisat orbit 28825. Nighttime measurements are indicated by red crosses and daytime measurements by red plus signs at the bottom of the plots. White areas are missing data due to clouds, in particular PSCs near the South Pole. The left-hand panel is repeated from Fig.5b, but with a truncated vertical range.

in von Clarmann et al. (2015). Thereafter, the profiles are transformed to boxcar base functions with constant volume mixing ratios within a layer, which we assume to be more appropriate for comparison with model data. The layer boundaries are in-between the coarse grid pressure levels.

In Figure 7 we give an example of the ClO coarse grid representation (transformed to layers with constant VMRs) for
orbit 28825 of 4 September 2007. As reference, the left panel shows the standard retrieval result, which has already been discussed in Section 7.3. For better comparability the height region is restricted to 5–45 km here. The alternative coarse grid representation (right panel), where each tile represents one degree of freedom, shows the same ClO maximum in the southern polar stratosphere as the standard retrieval and very similar, noisy patterns of nearly ClO-free regions in the rest of the lower stratosphere, as well as moderately enhanced ClO in the upper stratosphere.

**8   MIPAS ClO climatology**

### 8.1   Time series of V5 and V8 ClO

Figure 8 shows time series of monthly daytime averages of MIPAS V5 (left column) and V8 (right column) ClO data at the altitudes of 40, 30, 25, 20 and 15 km (top to bottom). Here and in the following we define the boundary between day and night by a solar zenith angle (SZA) of 94°, for which the Sun is just still visible at 20 km above the Earth's surface (light
diffraction neglected). At the altitudes of 15–25 km, but most pronounced at 20 km, there are ClO enhancements in Arctic winter and spring, and a factor of two to three stronger enhancements in Antarctic winter and spring, while during the other seasons and at lower latitudes almost no ClO was observed. As a consequence of the spectroscopic update from HITRAN-1996 to HITRAN-2016, the V8 ClO maxima at 20 km are about 10% higher than those of the V5 data set. This is more visible in





Fig. 9, in which curves at 20 km for the latitude band 60°S–90°S are presented. Because of other updates in the V8 retrievals (weaker regularization above 20 km, modified retrieval of continuum and offset), the ratio between the V8 and V5 maxima is even larger at 25 km. In Fig. 8, interannual variations of the ClO maxima are difficult to detect, although at 20 km the somewhat higher Arctic ClO amounts in the winters/springs of 2004/2005 and 2010/2011 and a nearly completely missing Arctic ClO

enhancement during January to March 2004 can be noticed. More details about interannual variations in lower stratospheric Arctic and Antarctic ClO will be discussed in Sections 8.4, 9.2 and 9.3. At 15 and 20 km the background values of both data sets are close to zero. While the V5 data set at these altitudes contains slightly negative zonal mean ClO values of up to -0.1 ppbv, the V8 data are consistently around zero at 15 km or slightly positive (up to 0.1 ppbv) at 20 km. Further, there is a continuous decrease of the V5 VMRs at 25 and especially at 30 km after 2006. In contrast, the V8 background values at these

levels slightly increase from the FR to the RR period, but show no obvious trends thereafter. In addition, the V8 ClO VMRs at 25 and 30 km are 0.1–0.2 ppbv higher than the V5 values. A reason for the greater time stability of the V8 data might be the better spectral calibration, taking detector aging into account (Kleinert et al., 2018), of the V8 spectra applied for retrieval.

The largest differences between the V5 and V8 data occur in the altitude region of the upper stratospheric maximum at 40 km. Concerning the V5 retrievals, at this altitude only the ClO amounts of the FR period (July 2002–March 2004) are in

a reasonable range of 0.4–0.7 ppbv. The VMRs resulting from the V5 retrievals of the RR data set are considerably higher, namely up to 1.0 ppbv after the beginning of 2005 and up to 1.3 ppbv after September 2006 over nearly all latitudes, which is about twice as high as the upper stratospheric maxima observed by MLS (Froidevaux et al., 2022) or by SMILES on the International Space Station (Sato et al., 2012). Due to improved modeling of the atmospheric continuum and of the instrumental offset as outlined above, the upper stratospheric maxima of the V8 data set are continuously between 0.4 and 0.6 ppbv for the

FR as well as for the RR period. Both at northern and at southern mid-to-high latitudes there are strong seasonal variations, with maxima in hemispheric summer and minima in hemispheric winter. While the northern hemispheric maxima are about 0.6 ppbv, the southern hemispheric maxima are somewhat weaker, around 0.5 ppbv. The VMRs observed in the tropics are around 0.4 ppbv and exhibit weaker seasonal variations. Especially in the southern hemisphere, a latitudinal shift of the position of these maxima is visible, namely from July at the Equator to January at 50°S. Much of the upper stratospheric variability of ClO

can be explained by changes in $CH_4$, because the reaction $CH_4 + Cl \rightarrow CH_3 + HCl$ converts reactive chlorine (Cl + ClO) into the reservoir species HCl (Solomon and Garcia, 1984; Aellig et al., 1996; Froidevaux et al., 2000). Thus, the increase of ClO at 40 km towards high latitudes during summer is caused by a corresponding decrease of $CH_4$, that results from the persistent descent of mesospheric air, which contains low $CH_4$ abundances, during and after the preceding winter (Siskind et al., 2016). During winter, this process is balanced by the slow production of ClO from all chlorine reservoir gases (HCl, $ClONO_2$, and

HOCl) due to the lack of sunlight (Aellig et al., 1996). Basically, the V8 distributions at 40 km altitude are rather similar for all years. The most apparent interannual variation is the weaker ClO maximum in the northern extra-tropics in 2011. As shown by Siskind et al. (2016), this anomaly was caused by relatively weak wintertime descent in the Arctic vortex followed by strong springtime planetary wave activity, leading to unusually high summertime $CH_4$ amounts.





**Figure 8.** Left: Time series of monthly daytime (solar zenith angle $< 94°$) averages of MIPAS V5 ClO data, measured at **(a)** 40 km, **(c)** 30 km, **(e)** 25 km, **(g)** 20 km, and **(i)** 15 km altitude. Right: Same layout as in the left-hand panel, but for MIPAS V8 ClO data. The white areas at high latitudes are data gaps due to polar night. ClO VMRs below or above the colour scale are set to the lowest or highest colour value, respectively. Note the extended VMR scale in Figs. 8a,g,h.



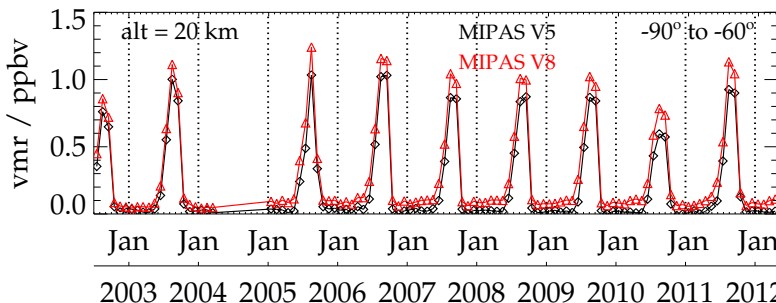

**Figure 9.** Time series of monthly daytime (solar zenith angle $< 94°$) averages of V5 (black) and V8 (red) MIPAS ClO data, measured at 20 km altitude in the latitude band $60°$–$90°$S.

## 8.2 Zonal distributions

Figure 10 shows climatological zonal distributions of daytime V8 ClO for December, January, February, March, and April (left column) and for June, July, August, September, and October (right column) of the entire MIPAS data set. Thus, in each row Arctic and Antarctic chlorine activation can be compared for corresponding times in the course of northern and southern hemi-

spheric winter and spring. Except in the polar regions during winter, for each of the presented months an upper stratospheric maximum of 0.5–0.6 ppbv is present around 37–40 km at all latitudes. In both hemispheres this maximum is somewhat more distinct and at slightly higher altitudes during summer. The reason for the low values at high latitudes during winter will be discussed in Sections 9.2 and 9.4.

Around 20 km altitude the buildup and disappeareance of the polar lower stratospheric ClO maxima caused by heterogeneous

chlorine activation in polar stratospheric clouds can be seen during winter/spring months. In the Arctic, a weak ClO maximum appears around $60°$N in December, which increases to about $\sim$0.5 ppbv in January and February and then decreases in March. In April, after the breakdown of the polar vortex in the climatological data set presented here, it has disappeared. The Antarctic ClO maximum observed in June is comparably as weak as its Arctic counterpart, but in July its spatial extent is already larger than the size of the respective Arctic maximum. In August and September, the Antarctic maximum increases considerably in

size as well as in magnitude (up to 1.2 ppbv), which is twice as large as the maximum observed in the Arctic. The reason is the stronger isolation of the air masses in the Antarctic vortex, leading to colder lower stratospheric temperatures and much more widespread PSC formation. In October the climatological Antarctic lower stratospheric ClO maximum has vanished as well. This climatological behavior (spatial and seasonal variation) is in qualitative good agreement with the ClO climatology measured by the MLS on the Upper Atmosphere Research Satellite (UARS) during the years 1991–1998 (Santee et al., 2003),

whereby it has to be considered that in Figs. 1 and 2 of this paper for each year ClO averages of the days with maximum chlorine activation are shown.





**Figure 10.** Left: Zonal distributions of MIPAS V8 daytime (SZA $< 94°$) ClO measurements in **(a)** December, **(c)** January, **(e)** February, **(g)** March, and **(i)** April. Right: Same as left, but for **(b)** June, **(d)** July, **(f)** August, **(h)** September, and **(j)** October. The distributions are averaged over the entire measurement period of MIPAS (2002–2012). White areas are missing daytime data due to polar night (high latitudes) or tropical regions not covered by the MIPAS scan.





**Figure 11.** Left: Zonal distributions of MIPAS V8 nighttime (SZA ≥ 94°) ClO measurements in **(a)** December, **(c)** January, **(e)** February, **(g)** March, and **(i)** April. Right: Same as left, but for **(b)** June, **(d)** July, **(f)** August, **(h)** September, and **(j)** October. The distributions are averaged over the entire measurement period of MIPAS (2002–2012). White areas are missing nighttime data due to polar day (high latitudes) or tropical regions not covered by the MIPAS scan. Note the reduced VMR scale as compared to Fig.10.





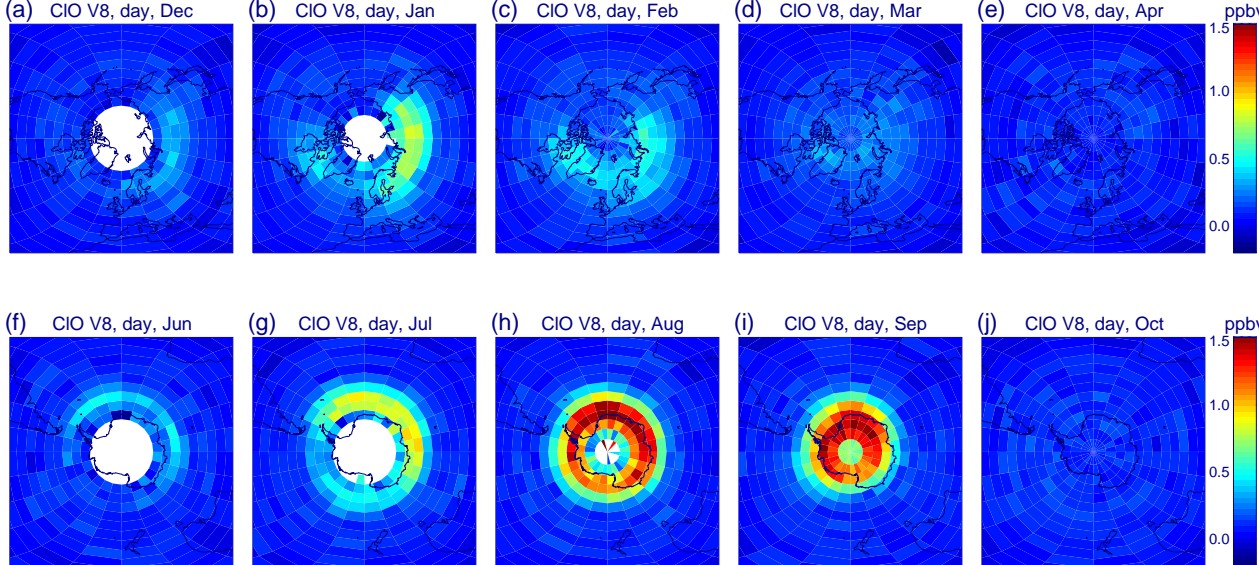

**Figure 12.** Top row: Northern hemispheric daytime (SZA $< 94°$) distributions of MIPAS V8 ClO at 20 km altitude, measured in **(a)** December, **(b)** January, **(c)** February, **(d)** March, and **(e)** April. The distributions are averaged over the full measurement period of MIPAS (2002–2012). Bottom row: Same as top row, but for southern hemispheric measurements in **(f)** June, **(g)** July, **(h)** August, **(i)** September, and **(j)** October. For white areas, no daytime observations are available.

The respective nighttime (SZA $\geq 94°$) zonal ClO distributions are displayed in Fig. 11. The measured ClO amounts are considerably lower than the daytime observations in the upper stratosphere, and the lower stratospheric maxima during polar winter are between 0.1 and 0.2 ppbv at the most. The reason for the diurnal variation in the upper stratosphere is nighttime transformation of ClO into the chlorine reservoirs $ClONO_2$ and HOCl (Ko and Sze, 1984). The diurnal variation in the polar winter lower stratosphere, which is generally strongly denoxified, is caused by transformation of ClO into its dimer ClOOCl during the night (Molina and Molina, 1987; Barrett et al., 1988; Cox and Hayman, 1988; Anderson et al., 1989).

### 8.3 Polar distributions at 20 km

Figure 12 shows northern (top row) and southern (bottom row) hemispheric distributions of daytime V8 ClO at 20 km altitude for the same climatological monthly composites as above, i.e., December to April and June to October, respectively. In the Arctic, weak ClO enhancements amounting to 0.3–0.4 ppbv are visible in December (Fig. 12a) around 60°N, mainly above the northern part of Russia extending as far as central Siberia. In January, the "hot spot" of enhanced ClO has extended in size, covering Scandinavia, northern Russia and Siberia, and exhibits VMRs of up to 0.8 ppbv. Further, there is a second, but weaker, maximum between Greenland and Arctic Canada. In February, the eastern hemispheric ClO enhancement has decreased in magnitude, and in March a considerable further decrease has occurred in the eastern as well as in the western hemisphere. In




April, differently to the zonal averages presented in Figure 10i, there are some weak remains of slightly enhanced ClO amounts (∼0.2 ppbv) scattered over wider latitudes.

In the southern hemisphere, the ClO enhancement in June (Fig. 12f) is somewhat stronger than the NH enhancement in December and covers a larger longitudinal area. The same applies for the southern hemispheric ClO enhancement in July (Fig. 12g) as compared to the northern hemisphere in January (Fig. 12b). In this month, ClO amounts of 0.8 ppbv were measured over a wide longitudinal region. In August and September, much higher ClO VMRs (1.2–1.5 ppbv) than in the respective NH months were observed in a circumpolar belt. With sunlight coming back to the polar region, this belt shifts towards higher latitudes in September. In this longitudinally resolved display, there are still slightly enhanced ClO amounts in October in the Antarctic and at southern mid-latitudes, somewhat more uniformly distributed than in the Arctic during April. Both for the northern and southern hemisphere, this pole-centered display of MIPAS ClO is also in qualitatively good agreement with the ClO distribution observed by MLS on UARS (Santee et al., 2003).

## 8.4 Interannual variations of ten-day polar averages at 20 km

For a closer comparison of interannual variations in polar chlorine activation, Fig. 13 shows annual time series of daytime ten-day averages of V8 ClO at 20 km altitude in the Arctic ($60°$–$90°$N) and Antarctic ($60°$–$90°$S) regions.

During every year, Arctic ClO (Fig. 13a) generally reveals very low background values between 0.05 and 0.1 ppbv from April to November. The ClO enhancement starts in December and at the latest ends by the end of March. The winters of 2003, 2007–2010 and 2012 are characterized by ClO maxima of 0.4–0.6 ppbv during January and/or February, but the winters of 2004, 2005 and 2011 show a different picture: In the year 2004 no ClO enhancements are visible, while in the Arctic winter of 2005 there is a ClO maximum as high as 1.0 ppbv at the beginning of February. However, this high value is caused by a low measurement rate of MIPAS during the first half of February 2005, when by happenstance only the chlorine-activated part of the Arctic vortex was sounded. The year 2011 is characterized by moderately enhanced ClO during February, followed by a strong, unusually late ClO enhancement during March. Detailed investigations of the Arctic winter of 2010/2011, with extraordinarily long-lasting cold conditions in the lower stratosphere and unprecedented ozone loss, have been performed by several authors (e.g. Manney et al., 2011; Sinnhuber et al., 2011; Arnone et al., 2012). The latter authors employed the official ESA product of MIPAS ClO, $O_3$ and other gases.

Antarctic ClO (Fig. 13b) also exhibits background values between 0.05 and 0.1 ppbv from mid-October until mid-May. The ClO amounts start to increase at the beginning of June, attain maximum values of 0.8–1.2 ppbv around the end of August/early September, and then decrease during September and the beginning of October. During June to August 2005 there are temporal oscillations in the ClO VMRs. These oscillations are also a sampling effect caused by various days with an incomplete number of measurements during this Antarctic winter, covering either chlorine activated or not activated parts of the vortex. Further, the years 2002 and 2010 are characterized by clearly lower ClO maxima than observed in the other years. The reasons for the low ClO amounts observed in 2002 and 2010 are unusual Antarctic winters. In the year 2002, three minor Sudden Stratospheric Warmings (SSWs) occurred during August and September, followed by a major Stratospheric Warming at the end of September,





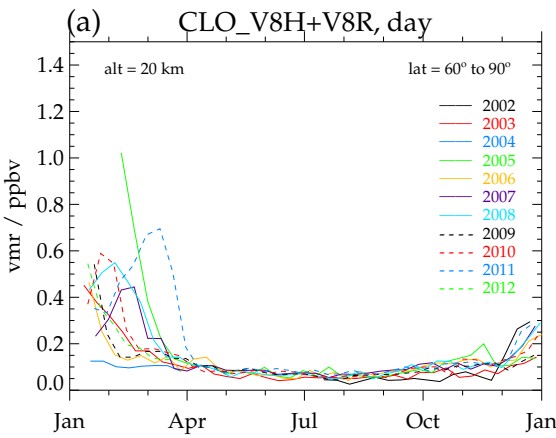 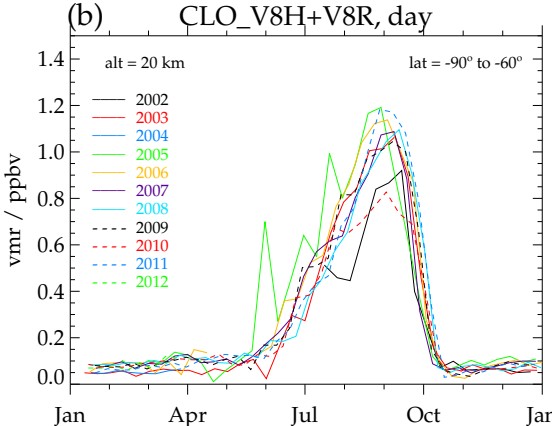

**Figure 13.** Time series of daytime (SZA < 94°) ten-day averages of MIPAS V8 ClO at 20 km altitude in the latitude bands **(a)** 60°N–90°N and **(b)** 60°S–90°S, measured during the years 2002–2012.

which caused a vortex split around 25 September (see, e.g., Varotsos, 2002; Glatthor et al., 2005). Similar to 2002, the year 2010 was also characterized by a series of minor SSWs (de Laat et al., 2011).

## 9   Comparison with MLS data

In this section we compare MIPAS V8 with MLS v5.0 ClO data. In doing so we use monthly and ten-day averages of collocated
data in different latitude bands. The collocation criteria are a maximum temporal difference of 6 h and a maximum spatial distance of 500 km. MLS ClO data that do not meet the quality and convergence criteria given in the "Version 5.0x Level 2 and 3 data quality and description document" (Livesey et al., 2022) are not taken into account. The vertical resolution of MLS ClO is 3–4.5 km over the entire range of scientifically useful data (147–1 hPa) (Santee et al., 2008; Livesey et al., 2022), which is comparable to the vertical resolution of MIPAS in the lower stratosphere, but two to three times better in the upper stratosphere.
The single-profile precision of MLS ClO retrievals is ∼0.1 ppbv (1-$\sigma$ uncertainty) in the pressure range from 100 to 1.5 hPa (Livesey et al., 2022), which is also better than the precision of MIPAS ClO retrievals.

### 9.1   Zonal distributions of February and August 2007

Figure 14 shows zonal daytime ClO distributions, determined from collocated observations of MIPAS (top row) and MLS (middle row), and the corresponding differences (bottom row). The left column depicts measurements of February 2007 and
the right column of August 2007. In good agreement, both distributions exhibit a ClO maximum of about 0.5 ppbv around 70° N in the Arctic lower stratosphere during February (Fig. 14a,b). The ClO maxima around 70° S in the Antarctic lower stratosphere during August (Fig. 14d,e) are considerably stronger, up to 1.2 ppbv for MIPAS and 1.4 ppbv for MLS, and also in rather good agreement. However, the MLS maximum is centered at a somewhat lower pressure level, which leads to

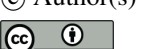



**Figure 14.** Left: Zonal distributions of collocated daytime (SZA < 94°) **(a)** MIPAS V8 and **(b)** MLS v5.0 ClO VMRs, measured in February 2007, and **(c)** of the MIPAS-MLS differences. Right: Same as left, but for August 2007. The collocation criteria are a maximum temporal difference of 6 h and a maximum spatial distance of 500 km.




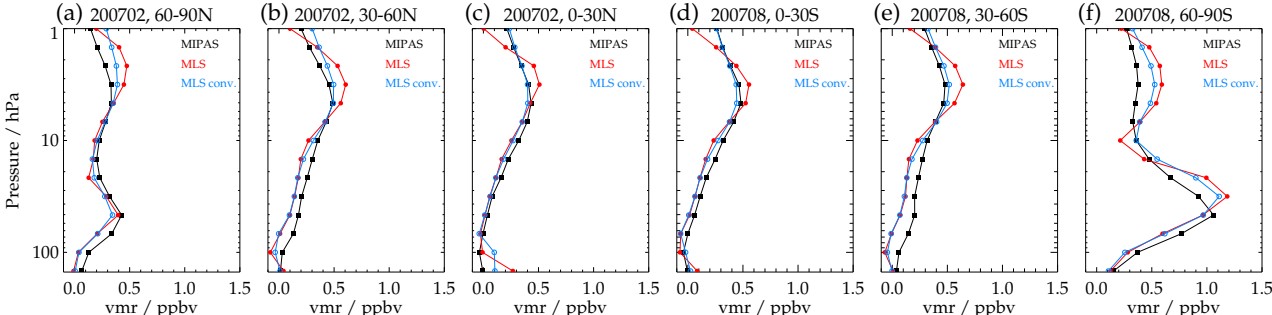

**Figure 15.** MIPAS (black) and MLS (red) daytime ClO profiles for February and August 2007 averaged over selected 30°-wide latitude bands (see plot titles). The blue curves represent the MLS profiles after convolution with a typical MIPAS averaging kernel.

MIPAS-MLS differences of up to -0.6 ppbv in this region (Fig. 14f). There is also fairly good agreement between the upper stratospheric maxima at 2–3 hPa at low-to-mid latitudes, and at high latitudes during summer. Larger differences of up to -0.4 ppbv are visible at high latitudes during winter (Fig. 14c,f), where MLS also observes upper stratospheric maxima while MIPAS measures much lower values. The reason for these differences will be investigated in Sections 9.2 and 9.4. Outside of the Antarctic vortex, the differences are between 0 and 0.2 ppbv in the lower and middle stratosphere. Due to the lower vertical resolution and larger measurement noise, the upper stratospheric maxima are generally a bit noisier in the MIPAS observations.

Inspection of MIPAS and MLS profiles of February and August 2007, averaged over 30°-wide latitude bands, enables a somewhat more quantitative comparison (Fig. 15). To account for the lower vertical resolution of MIPAS ClO, in particular in the upper stratosphere, MLS profiles convolved with a typical MIPAS averaging kernel are also shown. However, since the vertical resolution of MLS is only up to a factor of two to three better than that of MIPAS, convolution of the MLS profiles with a MIPAS averaging kernel tends to a somewhat too strong smoothing, i.e. to an upper approximation of the adjustment. As stated above, there is good agreement between the Arctic lower stratospheric maxima in February 2007 (Fig. 15a). The Antarctic lower stratospheric maximum of MLS in August 2007 is about 10% stronger and situated at a level that is 15 hPa lower in pressure (i.e., at higher altitude) than the maximum of MIPAS (Fig. 15f). The upper stratospheric maximum of MLS exhibits higher VMRs than that of MIPAS for every latitude band presented, and for the February averages (Fig. 15a-c) it is situated at a 1–2 hPa lower pressure level. The largest differences in the upper stratosphere occur above the Antarctic, followed by the southern mid-latitudes and the Arctic. Convolution of the MLS profiles with the MIPAS averaging kernel leads to nearly perfect agreement at mid-latitudes and in the tropics. Taking the above-mentioned caveat with respect to the convolution into account, the bias between the Arctic upper stratospheric maxima observed in February is reduced by up to 50% and that between the Antarctic maxima observed in August by up to 30%. Thus, these differences cannot be explained by the different vertical resolution of the two data sets alone. As mentioned above, these differences will be further investigated in Sections 9.2 and 9.4.





## 9.2 Time series of monthly averages at 31.62 and 3.16 hPa

Figure 16 shows time series of monthly averages of daytime ClO observed by MIPAS and by MLS between January 2005 and April 2012 in 30°-wide latitude bands at the pressure levels of 31.62 and 3.16 hPa, considering the above-mentioned collocation criteria. These pressure levels belong to the MLS retrieval grid and are in the altitude regions of the lower and

upper stratospheric ClO maxima. Time series of MLS ClO convolved with the same MIPAS averaging kernel as applied above are also shown. Apart from a slight bias of 0.05–0.1 ppbv in the background values, there is good agreement between MIPAS and MLS ClO at the pressure level of 31.62 hPa (left column). In the Arctic lower stratosphere (Fig. 16a), both instruments observe ClO maxima of about 0.5 ppbv during January/February, and the same interannual variations. While the maxima are of equal strength in 2007, 2009 and 2012, the MLS maxima are up to 0.15 ppbv higher than those of MIPAS in the other years.

In the Antarctic lower stratosphere (Fig. 16k), both instruments in good temporal agreement observe ClO maxima of the order of 1 ppbv during August/September, with the MLS values being generally higher, on average by 16%. In both data sets the year 2010 stands out with an unusually weak maximum, caused by a series of minor SSWs (de Laat et al., 2011). During every year, the largest Antarctic differences occur during July, with MLS values of up to 0.4 ppbv above the MIPAS VMRs. The ClO time series in the mid-latitude bands 30°–60°N and 30°–60°S of both instruments are correlated with the respective polar time

series, which reflects chlorine activation in the poleward parts of these bands. A better separation between pure mid-latitude ClO measurements and observations from inside the polar vortices could be achieved by display of the data against equivalent latitude and filtering with respect to potential vorticity, but this is beyond the focus of this investigation. In contrast to the polar regions, the midlatitude ClO maxima observed by MLS are equal to or lower than the MIPAS maxima. In the tropics and subtropics (0°–30°N and 0°–30°S), the retrieved ClO abundances are very low for both instruments throughout the year and

comparable to the minimum values at mid-latitudes, namely around 0.05 ppbv for MLS and 0.1 ppbv for MIPAS.

At 31.62 hPa, convolution of the MLS data with a MIPAS averaging kernel leads to somewhat smaller polar ClO maxima and thus to even better agreement with the MIPAS data. However, the differences between Antarctic MIPAS and MLS ClO during July are hardly reduced. In the other latitude regions, convolution has little influence on the MLS data at this pressure level.

At the pressure level of 3.16 hPa (right column), the time series of the two instruments agree fairly well in the Arctic and in the northern and southern tropics. In the Arctic, both curves show similar seasonal variations, with broad maxima of 0.5–0.65 ppbv peaking in boreal summer and minima in boreal winter. However, the minima in MIPAS ClO are about 0.1 ppbv lower. The processes leading to the seasonal variation have been discussed in Section 8.1. At northern mid-latitudes MLS ClO exhibits almost no seasonal variations, with VMRs around 0.6 ppbv. In contrast, MIPAS ClO in this latitude band shows similar but

attenuated seasonal variations as in the Arctic latitude band, leading to 0.1–0.2 ppbv lower values than MLS during winter.

In the northern tropics neither data set shows substantial systematic variations, while in the southern tropics they show seasonal variations with maxima during July to September and minima during February to April. Larger differences between the two data sets occur at southern mid-to-high latitudes. As in the northern hemisphere, the MIPAS time series at southern mid-latitudes exhibits clear seasonal variations, with maxima of about 0.5 ppbv during austral summer and minima of around





**Figure 16.** Time series of monthly daytime (SZA < 94°) averages of collocated MIPAS V8 (black) and MLS v5.0 (red) ClO, measured during January 2005 to April 2012 at 31.62 hPa (left) and at 3.16 hPa (right) in the latitude bands **(a, b)** 60°–90°N, **(c, d)** 30°–60°N, **(e, f)** 0°–30°N, **(g, h)** 0°–30°S, **(i, j)** 30°–60°S, and **(k, l)** 60°–90°S. The blue curves represent MLS ClO convolved with a typical MIPAS averaging kernel. The collocation criteria are a maximum temporal difference of 6 h and a maximum spatial distance of 500 km. Note the varying VMR-scales.





0.3 ppbv during austral winter. These minima do not appear in the MLS time series, which is characterized by weaker seasonal variations and temporally more restricted maxima in austral spring. The largest differences are visible in the Antarctic latitude band, where MIPAS ClO exhibits broad maxima around 0.5 ppbv between September and March and distinct minima of about 0.1–0.2 ppbv during June/July, i.e. similar seasonal variations as in the Arctic. The MLS data show much weaker and semi-
annual variations, with maxima in April and September and minima in December and June/July - the latter around 0.5 ppbv, i.e. 0.2–0.3 ppbv higher than the respective minima in MIPAS ClO.

At 3.16 hPa, convolution with the MIPAS averaging kernel leads to better agreement at northern mid-latitudes and to nearly perfect consistency in the tropics. In the Arctic, the convolution causes somewhat lower MLS values and worse agreement in the summer half-year, but almost no changes during winter. The differences between MIPAS and MLS at southern mid-latitudes
are scaled down by about 30%, but those in the Antarctic latitude band are hardly reduced after convolution.

A potential reason for the substantial differences between the ClO values of MIPAS and MLS in the Antarctic lower strato-sphere during July as well as in the southern mid-latitude and Antarctic upper stratosphere during austral winter is the larger difference in local time between the measurements of the two instruments at mid-to-high southern latitudes, which is caused by the different equatorial crossing times, the opposite flight directions and the turn of the viewing direction of MIPAS towards
the poles at latitudes higher than $80°$. A similar effect might also be responsible for the somewhat smaller differences in the Arctic during boreal winter. For an example check of this assumption we present EMAC simulations for ClO together with MIPAS and MLS data in Section 9.4.

## 9.3 Interannual variations of ten-day polar averages at 31.62 hPa

For a comparison of MIPAS and MLS ClO at higher temporal resolution, Fig. 17 shows daytime ten-day averages of collocated
observations at 31.62 hPa in the Arctic (top row) and Antarctic (bottom row) during the years 2006–2008 (left column) and 2009–2011 (right column). The displayed MIPAS data are rather similar to the observations at 20 km discussed in Section 8.4, but some differences arise from the somewhat higher geometrical altitude corresponding to the pressure level displayed here and from the applied collocation criteria. As in Fig. 16, the ClO background levels observed during the undisturbed times of the year are not much different, but the MIPAS data exhibit more scatter. In the Arctic, there is very good agreement between the
measurements of the two instruments during the period of chlorine activation for each of the years presented. In the Antarctic, there is generally rather good temporal agreement with respect to the maximum and the end of chlorine activation. In the onset phase during July, considerable differences with up to 0.5 ppbv higher MLS values exist, which will also be further analysed in Secion 9.4. Consistent with the monthly means (Fig. 16), the MLS maxima around the end of August exceed those of MIPAS by up to 0.2 ppbv.

## 9.4 Comparison with EMAC simulations

To check whether the differences between MIPAS and MLS ClO that occur at mid-to-high southern latitudes during austral winter and at high northern latitudes during boreal winter result from the large differences in local solar time of the measure-ments of the two instruments at these latitudes, we performed EMAC simulations for ClO and related species for 1–2 January



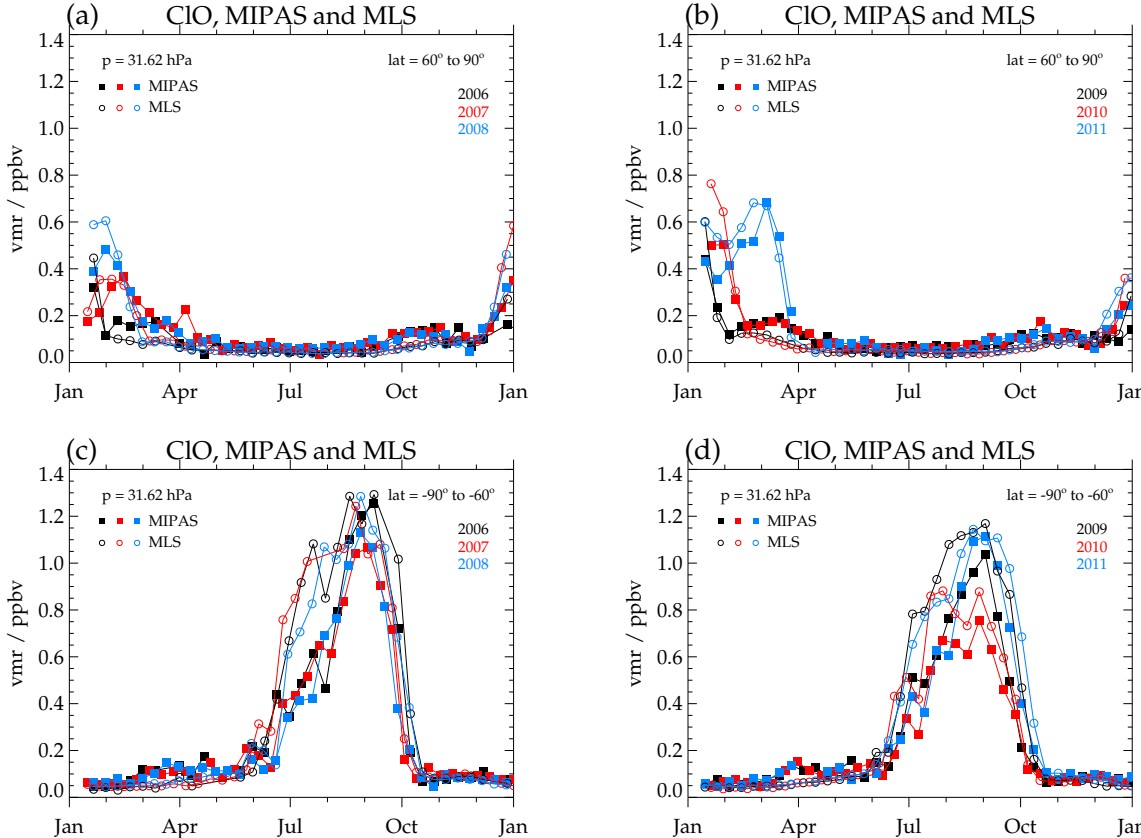

**Figure 17.** Top row: Time series of daytime (SZA < 94°) ten-day averages of collocated MIPAS V8 and MLS v5.0 ClO measured at 31.62 hPa in the latitude band 60°N–90°N during the years **(a)** 2006–2008 and **(b)** 2009–2011. Bottom row: Same as top row, but for the latitude band 60°S–90°S. The collocation criteria are a maximum temporal difference of 6 h and a maximum spatial distance of 500 km.

and 1–2 July 2005, with a model output in time steps of 720 s. In Figure 18 we show mean EMAC daytime ClO VMRs in 1-h-wide bands of local solar time, convolved with the same MIPAS averaging kernel as applied above. The curves represent averages over southern high latitudes at 31.62 hPa for 1–2 July 2005 (Fig. 18a), over northern high latitudes at 3.16 hPa for 1–2 January 2005 (Fig. 18b), and over southern mid-latitudes (Fig. 18c) and high latitudes (Fig. 18d) at 3.16 hPa for 1–2 July

5 2005. The EMAC ClO data are characterized by strong variations with local time, ranging from 0.3 to 0.85 ppbv at 31.62 hPa and from about 0.18 to 0.53 ppbv at 3.16 hPa. The respective mean MIPAS and MLS ClO VMRs, plotted against the average local solar time of their observation, are also specified. For statistical reasons these data are averaged over 1–2 January of the years 2005–2012 and over 1–2 July of the years 2005–2011, respectively. The MLS data are also convolved with the MIPAS averaging kernel. Based on the EMAC curves, the bias between Antarctic MLS and MIPAS ClO at 31.62 hPa during July,

10 which is around 0.3 ppbv in Fig. 16 and up to 0.5 ppbv in Fig.17, to a large extent is explained by the differences in local solar time. Further, the differences between MIPAS and MLS ClO at 3.16 hPa at northern high latitudes during boreal winter and at





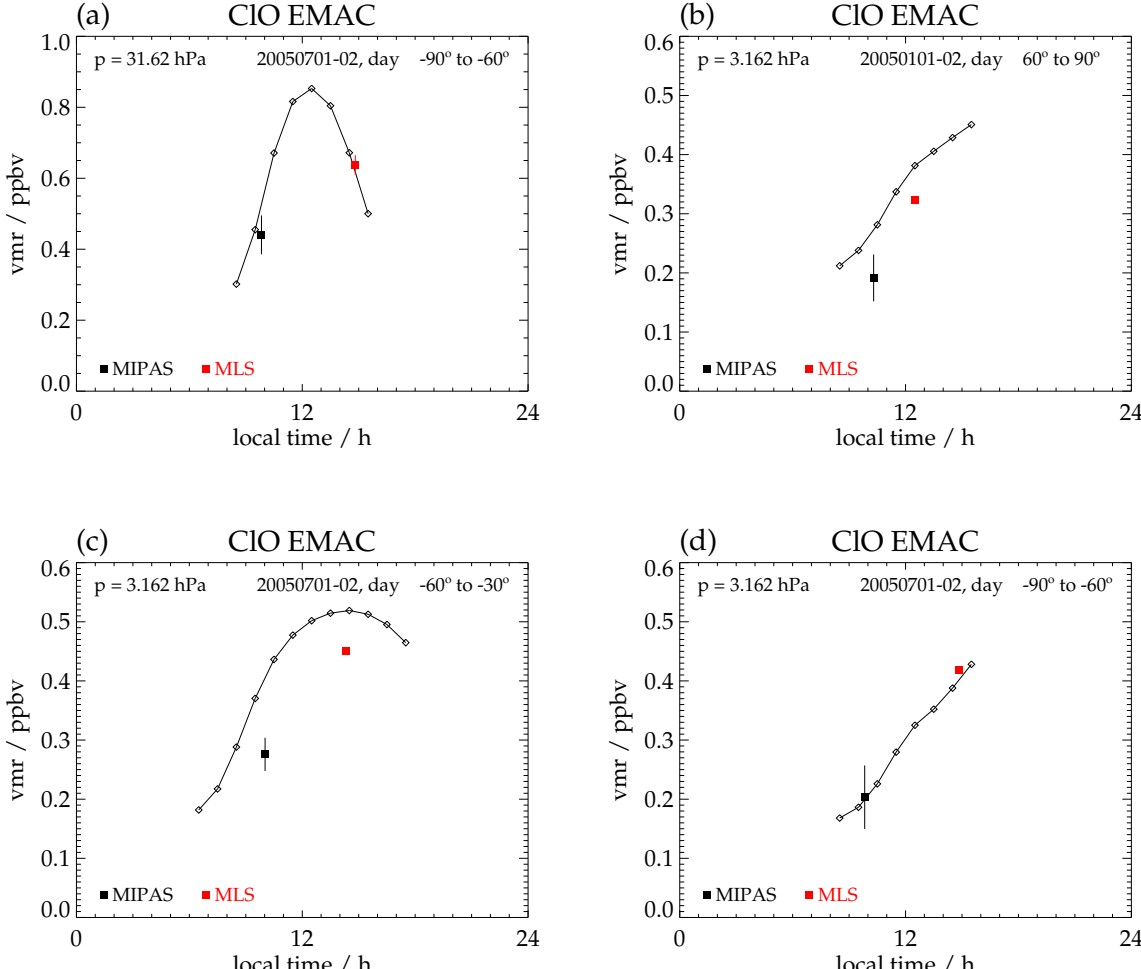

**Figure 18.** Simulated EMAC daytime (SZA < 94°) ClO volume mixing ratios versus local time for **(a)** 1–2 July 2005 at 31.62 hPa in the latitude band 60°S–90°S, **(b)** 1–2 January 2005 at 3.16 hPa in the latitude band 60°N–90°N, and for 1–2 July 2005 at 3.16 hPa in the latitude bands **(c)** 30°S–60°S and **(d)** 60°S–90°S. The black and red squares are MIPAS and MLS ClO VMRs averaged over the respective daytime measurements on 1–2 January 2005–2012 and on 1–2 July 2005–2011 in the respective latitude bands, plotted against the mean local solar time of the measurements. EMAC and MLS data are convolved with a MIPAS averaging kernel. The vertical lines denote the standard errors of the mean (SEMs) of the measurements. For the upper stratospheric MLS data these are within the symbol size.

southern mid-to-high latitudes during austral winter in Figs. 14 and 16 are well explained by the differences in local solar time. In particular, the MIPAS and MLS data observed during Antarctic winter perfectly fit to the EMAC curve (Fig. 18d).





## 10 Conclusions

A new ClO data set (versions V8H_CLO_62, V8R_CLO_162, V8R_CLO_262, V8R_CLO_562) has been retrieved from version 8 infrared limb emission spectra recorded with MIPAS on Envisat. The data set covers the entire operational period of MIPAS from July 2002 to April 2012. The update of the spectroscopy for ClO line modeling from HITRAN-1996 to HITRAN-

2016 resulted in about 10% higher ClO volume mixing ratios in the lower stratosphere than in the previous data version. In the middle and upper stratosphere there is an additive effect of the change in spectroscopy and the other updates, e.g., modeling of the offset and of the continuum. As in the previous data version, the retrieval was performed using lines of the vibro-rotational fundamental band of ClO around 844 cm$^{-1}$. However, more lines of the ClO P-branch were used over the whole MIPAS scan, and in addition the lines of the complete ClO R-branch were employed at altitudes above 31.5 km. The latter extension led to

considerably better vertical resolution in the upper stratosphere, now amounting to 7.5–9.5 km at 40 km altitude. The vertical resolution in the lower stratosphere is about 4 km.

A shortcoming of the previous ClO retrievals, especially of the reduced resolution measurement period, was much too high VMRs at the altitude of the upper stratospheric maximum at 37–40 km. Shifting the upper end of the continuum retrieval from 32 to 58 km and performing a more sophisticated offset retrieval resulted in smaller upper stratospheric VMRs for retrievals

of the FR and even more of the RR period, which now agree well with the retrieval results of the FR period and with MLS data. The retrieval error is 0.4–0.5 ppbv at 20 km and 0.7–0.8 ppbv at 40 km altitude. Accordingly, chlorine activation in the lower stratosphere under polar winter conditions (ClO VMRs of more than 1.5 ppbv) can be analysed using single MIPAS scans, but profile averaging has to be performed for analysis of the upper stratospheric ClO maximum (VMRs of ∼0.5 ppbv). Determination of horizontal information smearing and displacement shows that the horizontal resolution is sampling limited.

In addition to the standard data set, an alternative coarse grid representation has been provided to enable data averaging or comparison to model results without consideration of averaging kernels.

A comparison with collocated ClO measurements of the Microwave Limb Sounder on the Aura satellite shows good agreement between the magnitudes of the lower stratospheric maxima observed in polar winter and between the seasonal variations in the lower stratosphere. Somewhat larger differences occur in the Antarctic lower stratosphere during July, with

MLS values of up to 0.5 ppbv above those of MIPAS. Except in the southern hemispheric middle and high latitudes during austral winter, where the MIPAS VMRs are systematically lower than those of MLS, there is also fairly good agreement between the upper stratospheric maxima observed during the course of the year. For the most part, the lower and upper stratospheric differences are attributable to the differences in local solar time of the MLS and MIPAS measurements, as demonstrated through simulations with the EMAC model.

*Data availability.* MIPAS ClO data can be downloaded from the IMK data server (https://www.imk-asf.kit.edu/english/308.php).

*Supplement.* The supplement related to this article is available online at ....



*Author contributions.* NG has the final editorial responsibility for this paper, performed the test calculations needed to improve the retrieval setup, using suggestions from the entire group. He was also responsible for spectroscopy issues, coded and maintained parts of the error estimation software and provided most of the plots. The preparation of the manuscript was started by TvC and finished by NG. TvC organized related discussions and cared about TUNER compliance of error estimates. UG provided and maintained the retrieval software and coded and maintained parts of the error estimation software. SK and ALinden performed the retrievals. SK provided the horizontal averaging kernels. MK coded and maintained parts of the error estimation software, performed the error estimation and organized the calculation of the horizontal averaging kernels. OK performed the EMAC simulations. GPS was engaged in quality control, evaluated deficiencies in earlier data versions, suggested necessary improvements and took the responsibility for some key decisions related to the retrieval setup. BF, MGC and MLP took care that the retrieval setup was developed in a way that inter-consistence with the retrieval setups of middle atmospheric measurement modes was maintained. MLS performed a critical review of the intercomparison between MIPAS and MLS and gave helpful comments on the whole manuscript. All authors contributed to the development of the retrieval setup, discussed the results, and contributed to the final text.

*Competing interests.* At least one of the (co-)authors is a member of the editorial board of Atmospheric Measurement Techniques. The authors have no other competing interests to declare.

*Acknowledgements.* MIPAS version 8.03 spectra used for this work were provided by the European Space Agency. We would like to thank the MIPAS Quality Working Group for enlightening discussions, Claus Zehner for invaluable support, and Guido Levrini for motivating us to develop an independent MIPAS data processor. The project on which this manuscript is based was funded by the Federal Ministry for Economic Affairs under the grant number 50EE1547 (SEREMISA). The responsibility for the content of this publication is the responsibility of the authors. The computations were done in the frame of a Bundesprojekt (grant MIPAS_V7) on the Cray XC40 "Hazel Hen" of the High-Performance Computing Center Stuttgart (HLRS) of the University of Stuttgart. The IAA team acknowledges financial support from the Agencia Estatal de Investigación of the Ministerio de Ciencia, Innovación y Universidades through the project PID2022-141216NB-I00, as well as the Centre of Excellence "Severo Ochoa" award to the Instituto de Astrofísica de Andalucía (CEX2021-001131-S). Work at the Jet Propulsion Laboratory, California Institute of Technology, was carried out under a contract with the National Aeronautics and Space Administration (80NM0018D0004).



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
