# Peer review of "Version 8 IMK/IAA MIPAS measurements of ClO"

_EGUsphere, 2025_

## Referee Comment (RC1)

This article presents a new V8 data product for ClO from MIPAS.  It looks like a good data product, a significant improvement compared to the previous processing version for the altitude region near 40 km, i.e., in the vicinity of the stratospheric peak that is not linked to chlorine processing in the polar vortex.

Overall, I see no major problems with the manuscript.  There was just one place where it seemed an explanation may not have been completely explored.  There were differences observed between MLS and MIPAS ClO results in July, as shown in Figure 17 (panel c for Figure 17 is reproduced below, with an arrow indicating the largest discrepancies).

[Figure]

Model calculations were employed to show that discrepancy might be attributed to the difference in local time between the two instruments, as shown in Figure 18a, which is reproduced below:

[Figure]

This argument appears to be well supported, but to make it more complete, a similar calculation should be done for September 1st, where the ClO peaks and the differences between the two instruments (as seen in Figure 17c) are less pronounced. If the model calculations predict a smaller discrepancy for ClO at the two local times in September, that would add more weight to the argument. If the model calculations predict a similar difference at the two local times in September, that would make the argument more tenuous.

I will point out a couple of observations of the ClO data that need not be addressed for this manuscript but may serve as food for thought should there be a future processing version.

Figure 9 is reproduced below:

[Figure]

Full resolution                                                              Reduced resolution

   In the V8 results, there is a distinct step (increase) in the retrieved background ClO level after the instrument was switched to the reduced resolution mode (i.e., 2005 and later).  There also appears to be a persistent slope to the data during background periods between polar winter events for the reduced resolution period, but there is no hint of a slope for the full resolution period.  That suggests a possible artifact in the retrieval that is significant only for the reduced resolution period.  There is no apparent discrepancy between full resolution and reduced resolution in the V5 results.

[Figure]

   A portion of Figure 15 is reproduced above.  The arrows indicate the stratospheric ClO peak seen by MLS that MIPAS never seems to fully capture.  This is perhaps more evident in the difference plot between MIPAS and MLS from Figure 14, reproduced below:

[Figure]

MIPAS is persistently lower around 2 to 3 hPa and persistently higher near 1 hPa, which suggests the ClO retrieval is smearing the peak's contribution in altitude.  Since this is presumably associated with the altitude resolution of your retrieval around 1-3 hPa, I'm not sure if there is anything that can be done to improve the situation, but I thought I would mention the issue in case there was.

The systematic blue feature at the bottom of the above plot appears to be associated with enhanced tropical ClO in the MLS data, which I am not convinced is real, so not a problem in the MIPAS results.
* * *
**Minor comments:**

> The caption to Figure 1a mentions a green dashed line, but the only panel that features a green dashed line is Figure 1b.

> Page 14, line 2: internal line shape (ILS)

Do you not mean "instrumental line shape," defined as ILS in the footnotes to Table 5?

>In the titles for Figures 13a, 13b, 14b, and 14c: CLO

Should be ClO, without the capital "L"

---

## Author Comment (AC1)

Date: 15 October 2025

**Reply to the comments of reviewer #1 on the paper "Version 8 IMK/IAA MIPAS measurements of ClO", egusphere-2025-3352**

Norbert Glatthor et al.

15

Reviewer comments are in black, while our replies are in blue.

This article presents a new V8 data product for ClO from MIPAS. It looks like a good data product, a significant improvement compared to the previous processing version for the altitude region near 40 km, i.e., in the vicinity of the stratospheric peak that is not linked to chlorine processing in the polar vortex.

Overall, I see no major problems with the manuscript.

We thank Chris Boone for this positive assessment.

There was just one place where it seemed an explanation may not have been completely explored. There were differences observed between MLS and MIPAS CIO results in July, as shown in Figure 17 (panel c for Figure 17 is reproduced below, with an arrow indicating the largest discrepancies).

Model calculations were employed to show that discrepancy might be attributed to the difference in local time between the two instruments, as shown in Figure 18a, which is reproduced below:

This argument appears to be well supported, but to make it more complete, a similar calculation should be done for September 1st, where the ClO peaks and the differences between the two instruments (as seen in Figure 17c) are less pronounced. If the model calculations predict a smaller discrepancy for ClO at the two local times in September, that would add more weight to the argument. If the model calculations predict a similar difference at the two local times in September, that would make the argument more tenuous.

As suggested, we performed additional EMAC model calculations for 1-2 September 2005. The outcome is that - in contrast to the model results for 1-2 July - the EMAC calculations for 60–90S and 31.62 hPa predict nearly the same ClO amounts for the local times of the MIPAS and the MLS measurements (see Fig. 1 below). This is in agreement with the smaller differences between the two instruments in September as compared to July. The average ClO amounts observed by MIPAS and MLS on 1-2 September 2005–2011 are also shown. Due to convolution with a MIPAS averaging kernel, the MLS value in this display is even slightly lower than the MIPAS value. We will add the sentences

**Figure 1.** Simulated EMAC daytime (SZA  $< 94^{\circ}$ ) CIO volume mixing ratios versus local time for 1–2 September, 2005, at 31.62 hPa in the latitude band  $60^{\circ}$ S- $90^{\circ}$ S. The black and red squares are MIPAS and MLS CIO VMRs averaged over daytime measurements of 1–2 September 2005–2012 in the respective latitude band, plotted against the mean local solar time of the measurements. EMAC and MLS data are convolved with a MIPAS averaging kernel. The vertical lines denote the standard errors of the mean (SEMs) of the measurements.

"As a cross-check, we performed additional model calculations for 1-2 September 2005 (not shown). For the pressure level of 31.62 hPa, these calculations result in nearly the same Antarctic EMAC ClO VMRs for the local solar times of the MIPAS and of the MLS measurements, which corroborates the smaller differences between the two instruments in September as compared to July (see Fig. 17c,d)." at the end of Section 9.4 (P. 33, L. 2). However, because of the large amount of Figures in the current manuscript, we would rather abstain from adding the September results.

I will point out a couple of observations of the ClO data that need not be addressed for this manuscript but may serve as food for thought should there be a future processing version. Figure 9 is reproduced below:

In the V8 results, there is a distinct step (increase) in the retrieved background ClO level after the instrument was switched to the reduced resolution mode (i.e., 2005 and later). There also appears to be a persistent slope to the data during background periods between polar winter events for the reduced resolution period, but there is no hint of a slope for the full resolution period. That suggests a possible artifact in the retrieval that is significant only for the reduced resolution period. There is no apparent discrepancy between full resolution and reduced resolution in the V5 results.

At the moment, we do not have an explanation for the step in the background level and for the slope in retrieved V8 ClO of the reduced resolution period. However, both the step as well as the persistent slope are rather small, about 0.05 ppbv only. These issues will be revisited should the

**MIPAS data processing algorithms be updated in the future.**

A portion of Figure 15 is reproduced above. The arrows indicate the stratospheric ClO peak seen by MLS that MIPAS never seems to fully capture. This is perhaps more evident in the difference plot between MIPAS and MLS from Figure 14, reproduced below:

MIPAS is persistently lower around 2 to 3 hPa and persistently higher near 1 hPa, which suggests the CIO retrieval is smearing the peak's contribution in altitude. Since this is presumably associated with the altitude resolution of your retrieval around 1-3 hPa, I am not sure if there is anything that can be done to improve the situation, but I thought I would mention the issue in case there was. The systematic blue feature at the bottom of the above plot appears to be associated with enhanced tropical CIO in the MLS data, which I am not convinced is real, so not a problem in the MIPAS results.

We also think that, compared to MLS, the MIPAS CIO retrieval is smearing the peak's contribution in altitude. This assumption is confirmed by the good agreement of the MIPAS profiles with the convolved MLS profiles in Fig. 15b-e. The differences persisting after convolution in Fig. 15a and 15f are caused by the differences in local solar time. We think, we should not try to improve the situation by weakening the constraint, because the retrieval error at this altitude is already about 100% (see Fig.4 in the manuscript) We will add the sentence

"In this display it is also clearly visible that the MIPAS CIO retrieval is smearing the upper stratospheric maximum in altitude."

after the sentence "... situated at a 1–2 hPa lower pressure level." on P. 28, L. 16.

**Minor comments:**

70

> The caption to Figure 1a mentions a green dashed line, but the only panel that features a green dashed line is Figure 1b.

The reviewer is right. We will shift the sentence "green dashed line: measurement noise in terms of noise equivalent spectral radiance (NESR)" to point (b) of the caption.

75

> Page 14, line 2: internal line shape (ILS) Do you not mean instrumental line shape," defined as ILS in the footnotes to Table 5?

The reviewer is right. We will change "internal line shape" into "instrumental line shape."

>In the titles for Figures 13a, 13b, 14b, and 14c: CLO Should be ClO, without the capital "L". The titles will be corrected accordingly.

---

## Author Comment (AC2)

**Reply to the comments of reviewer #2 on the paper "Version 8 IMK/IAA MIPAS measurements of ClO", egusphere-2025-3352**

Norbert Glatthor et al.

Reviewer comments are in black, while our replies are in blue.

Glatthor et al. is a nice manuscript introducing a new version of the 2002–2012 MIPAS CIO measurements. It is a very welcome contribution to the available data on stratospheric CIO, both in the upper stratosphere and in the lower stratosphere during the polar winter.

We thank reviewer #2 for this encouraging assessment.

Pg. 1, L. 28 – "These days, monitoring of stratospheric ClO from the ground is a routine activity".
Yes, NDACC microwave measurements of ClO are "routine" in the sense that they are available on
most days, but this sentence seems to imply that they are generally available at NDACC stations.
There are only 2 instruments measuring ClO from the ground, one at Mauna Kea and one at Scott Base.

The reviewer obviously means P. 2, L. 28. We will replace the passage

"These days, monitoring of stratospheric CIO from the ground is a routine activity, in particular by stations associated with the Network for Detection of Atmospheric Composition Change (NDACC) (e.g. Solomon et al., 1984; de Zafra et al., 1994; Raffalski et al., 1998; Solomon et al., 2000; Nedoluha et al., 2011, 2025). These measurements were complemented by observations within the framework of specific measurement campaigns, using ground-based (de Zafra et al., 1989), airborne (e.g. Crewell et al., 1994; Wehr et al., 1995) and balloon-borne (e.g. Stachnik et al., 1992, 1999; Wetzel et al., 2010; de Lange et al., 2012) platforms."

by

"Since the 1980s and 1990s measurements of CIO in the microwave region from the ground have been performed at several stations, which are now associated with the Network for Detection of Atmospheric Composition Change (NDACC) (e.g. Solomon et al., 1984; de Zafra et al., 1994; Raffalski et al., 1998; Solomon et al., 2000; Nedoluha et al., 2011, 2025). These measurements were complemented by observations within the framework of specific measurement campaigns, using ground-based (de Zafra et al., 1989), airborne (e.g. Crewell et al., 1994; Wehr et al., 1995) and balloon-borne (e.g. Stachnik et al., 1992, 1999; Wetzel et al., 2010; de Lange et al., 2012) platforms.

These days, routine monitoring of ClO is carried out by the NDACC microwave instruments at Mauna Kea, Hawaii, and Scott Base, Antarctica."

Pg 3. L. 8 – "For data users who prefer not to work with averaging kernels, we also provide the data on a coarse grid, where averaging kernels do not need to be applied". Does this mean that there are two different retrievals being performed, or just that a smoothed version of the retrieval is supplied?

As outlined in Section 7.5, we performed additional coarse grid ClO retrievals for the entire MIPAS data set. To make this clearer, we will change the passage

"... we also provide the data on a coarse grid, where averaging kernels do not need to be applied." on P. 3, L. 8 into

"... we also performed independent coarse grid retrievals, to which averaging kernels need not be applied. These retrievals will be exemplified in Sect. 7.5."

and the passage

"... we offer an alternative data product where the related averaging kernels are unity;" on P. 14, L. 16f into

"... we offer an alternative data product, obtained by a retrieval, in which the related averaging kernels are unity;"

Pg 6., L. 13 – "The O3 results from the combined CIO-O3 retrieval are discarded because they are deemed inferior to the standard V8 O3 results." This should obviously never be the case for optimally chosen parameters for a combined CIO-O3 retrieval, but perhaps a clarification of what is meant by "inferior" would help here. Is it just the case that occasionally the inclusion of noisy channels needed for the CIO retrieval (and not the standard O3 retrieval) cause a bad O3 retrieval? Or does the inclusion of CIO adversely affect almost all O3 retrievals?

The  $O_3$  results from the combined ClO- $O_3$  retrieval are deemed inferior to the standard V8  $O_3$  results, because these microwindows are optimized for ClO retrieval only. However, since a complete avoidance of  $O_3$  signatures in the ClO microwindows is not possible,  $O_3$  is jointly fitted. To make things better understandable, we will change the passage on P.6, L.9–14:

"Version 8 ozone, which is also available, is used as first guess and a priori of a combined ClO and  $O_3$  retrieval. This is because  $O_3$  interferences in the ClO analysis windows are so large that even minor spectroscopic inconsistencies between the lines used for the standard ozone retrieval and those used in the ClO windows could have a sizable effect on the ClO results. Further, potential weak effects of non-local thermodynamic equilibrium (non-LTE) in these  $O_3$  lines are also caught by this approach. The  $O_3$  results from the combined ClO- $O_3$  retrieval are discarded because they are deemed inferior to the standard V8  $O_3$  results."

into

45

"Because O3 interferences in the ClO analysis windows are so large that even minor spectroscopic inconsistencies between the lines used for the standard V8 ozone retrieval and those in the ClO windows could have a sizable effect on the ClO results, O3 is jointly fitted in the ClO retrieval. The standard V8 O3 retrieval product is used as first guess and a priori. By jointly fitting of O3, potentially small effects of non-local thermodynamic equilibrium (non-LTE) in these O3 lines are also captured. The jointly fitted O3 results are discarded, because they are deemed inferior to the standard V8 O3 results obtained in microwindows dedicated for O3 retrieval."

Figure 3 – What does the CIO profile used here look like besides being "strongly enhanced"?

Actually, some more information on the "strongly enhanced" CIO profile is given on P. 9, L. 29–31:

"... strongly enhanced ClO (2.65 ppbv) on 4 September 2007 at 21.3 km altitude in the Antarctic lower stratosphere (73.13°S, 162.09°W, solar elevation angle 6.67°)."

For more information in the caption of Fig. 3, we will change "... strongly enhanced ClO ..." into "... strongly enhanced ClO (2.65 ppbv) ...".

Also, it would be very helpful here if the ClO spectrum from Figure 1b could be plotted as a third panel with the same horizontal axis as Figures 3a and 3b.

We will add the ClO spectrum to Figure 3 and add the sentence

"The green line in Fig.3b indicates the spectral signature corresponding to the retrieved ClO profile, and corroborates that the improvements in RMS (blue versus red) are caused by the modelled ClO." after the sentence ending in "... is visible in the red negative spikes in the residual spectrum (Fig. 3b)." on P., L. 1/2.

Doesn't the fact that, wherever there is a clear red/blue difference the residual is still very negative imply that there is still not enough ClO in the model?

The residuals are dominated by spectral noise and are not significantly more negative at the spectral positions of the ClO lines. However, the red/blue differences in the residuals show that the retrieval without ClO (red) cannot compensate for the emission in the ClO lines, which are modelled when ClO is included in the retrieval (blue). This will probably become more obvious after addition of the ClO spectrum to Fig. 3, showing that the improvements in the adjustment (blue minus red spikes) are often nearly identical to the green ClO signatures.

95

Pg. 14, L. 17 – "where the related averaging kernels are unity; i.e. the profiles are free of formal a priori information". This can never be the case. "Near unity" and "contain minimal a priori information" would be okay.

100 Here we disagree with the reviewer. Unity averaging kernels can well be the case for an unconstrained retrieval free of apriori information, as performed in our coarse grid retrievals. In the CIO

Figure 1. Diagonal elements of the averaging kernels of the V8 ClO coarse grid retrieval along orbit 28825.

CGR we obtain unity averaging kernels in the height region of about 12–40 km (see Fig. 1 above). Therefore we do not intend to change the text in response to this comment.

105 It would be good to see these kernels as a fourth panel on Figure 4, which would also show the altitude range where these coarse resolution kernels are near unity.

We find it unfavourable to add additional coarse grid kernels to Fig. 4, because in the associated text in Section 7.1 we discuss the averaging kernels of the standard retrieval only. Instead, we suggest to add a third contour plot to Fig. 7, showing the diagonal elements of the CGR averaging kernels along orbit 28825 (see Fig. 1 above), and to discuss this plot with the sentences

"Figure 7c illustrates the diagonal elements of the averaging kernels (AKDs) of the ClO coarse grid retrieval. Except at the bottom of the retrieval range, the averaging kernels at each pressure level at and below 2 hPa have a peak value of 1."

at the end of Section 7.5 (P. 18, L. 9).

115

120

110

Pg. 31, L. 13 – The chemistry of the lower stratospheric peak and the upper stratospheric peak is completely different, so it's not clear from the data here whether the differences are larger because of a larger local time difference or to differences in the sensitivity to local time.

We do not discuss the differences between the lower and the upper stratospheric peaks here. In any case, the differences in local solar time of MIPAS and MLS measurements are larger at polar latitudes than at the Equator. For this reason we think that it is rather plausible that the differences between measurements of a tracer exhibiting a diurnal variation become larger at higher latitudes.

**We do not intend to change the text in response to this comment.**

- More generally, it would be very helpful somewhere earlier in the manuscript (perhaps on Pg. 26, if not earlier) to discuss the local times of both the MLS and MIPAS measurements, rather than to have the reader wondering about the cause of the bias between MLS and MIPAS shown in Figure 14. Currently it is not until the final figure (Figure 18) of the paper that the sensitivity of daytime measurements to local time is discussed.
- 130 The local solar times of the Equatorial crossing of MIPAS and MLS measurements are already specified in Section 2, and the reason for the increasing differences in local solar time are discussed on P. 31, L. 13–16. For more clarity earlier in the manuscript we will add the passage "Because of the opposite flight directions and the turn of the viewing direction of MIPAS towards the poles at latitudes higher than 80°, the differences in local solar time between MIPAS and MLS measurements at high latitudes become even larger, especially above the Antarctic."
  after P. 3, L. 31 in Section 2.
  - Pg. 33, last line "the MIPAS and MLS data observed during Antarctic winter perfectly fit to the EMAC curve". Since ClO in this region is dependent on the details of the presence of PSCs and therefore, when averaged over a large latitude band, on the precise variations in local temperature, the "perfect fit" in Figure 18a for this particular date seems serendipitous. Are the fits on other days during this period similarly good?
  - Actually we state a nearly perfect agreement for the data from 1–2 July at 3.16 hPa shown in Fig. 18d. At this pressure level PSCs do not occur. Unfortunately we do not have EMAC data for other days closely around 1–2 July. However, on request of reviewer #1 we performed additional EMAC model calculations for 1–2 September 2005. For this period, the Antarctic MIPAS and MLS CIO VMRs at 3.16 hPa also fit well to the EMAC curve (see Fig. 2 below). We will add the sentence "Further, these additional model calculations show that the smaller bias between Antarctic MIPAS and MLS CIO at 3.16 hPa during September can also well be explained by the difference in local solar time."

after the additions at the end of Section 9.4 (P. 33, L. 2) outlined in our response to reviewer #1.

150

**Figure 2.** Simulated EMAC daytime (SZA  $< 94^\circ$ ) CIO volume mixing ratios versus local time for 1–2 September, 2005, at 3.16 hPa in the latitude band  $60^\circ$ S– $90^\circ$ S. The black and red squares are MIPAS and MLS CIO VMRs averaged over daytime measurements of 1–2 September 2005–2012 in the respective latitude band, plotted against the mean local solar time of the measurements. EMAC and MLS data are convolved with a MIPAS averaging kernel. The vertical lines denote the standard errors of the mean (SEMs) of the measurements.

---

## Author Response (AR1)

**Rebuttal**

Reviewer comments and our replies are in black, while text changes and additions are in blue.

**Replies to the comments of reviewer #1, Chris Boone:**

**RC1** *This article presents a new V8 data product for ClO from MIPAS. It looks like a good data product, a significant improvement compared to the previous processing version for the altitude region near 40 km, i.e., in the vicinity of the stratospheric peak that is not linked to chlorine processing in the polar vortex.*

**Reply** We thank Chris Boone for this positive assessment.

**RC1** *Overall, I see no major problems with the manuscript. There was just one place where it seemed an explanation may not have been completely explored. There were differences observed between MLS and MIPAS ClO results in July, as shown in Figure 17 (panel c for Figure 17 is reproduced below, with an arrow indicating the largest discrepancies* (**see RC1 in the interactive discussion**)).
*Model calculations were employed to show that discrepancy might be attributed to the difference in local time between the two instruments, as shown in Figure 18a, which is reproduced below* (**see RC1 in the interactive discussion**)*:*
*This argument appears to be well supported, but to make it more complete, a similar calculation should be done for September 1st, where the ClO peaks and the differences between the two instruments (as seen in Figure 17c) are less pronounced. If the model calculations predict a smaller discrepancy for ClO at the two local times in September, that would add more weight to the argument. If the model calculations predict a similar difference at the two local times in September, that would make the argument more tenuous.*

**Reply** As suggested, we performed additional EMAC model calculations for 1-2 September 2005. The outcome is that - in contrast to the model results for 1–2 July - the EMAC calculations for 60–90S and 31.62 hPa predict nearly the same ClO amounts for the local times of the MIPAS and the MLS measurements (see Fig. 1 below). This is in agreement with the smaller differences between the two instruments in September as compared to July. The average ClO amounts observed by MIPAS and MLS on 1–2 September 2005–2011 are also shown. Due to convolution with a MIPAS averaging kernel, the MLS value in this display is even slightly lower than the MIPAS value. We added the sentences
"As a cross-check, we performed additional model calculations for 1-2 September 2005 (not shown). For the pressure level of 31.62 hPa, these calculations result in nearly the same Antarctic EMAC ClO VMRs for the local solar times of the MIPAS and of the MLS measurements, which corroborates the smaller differences between the two instruments in September as compared to July (see Fig. 17c,d)."
at the end of Section 9.4 (P. 33, L. 2). However, because of the large amount of Figures in the current manuscript, we abstained from adding the September results.

**RC1** *I will point out a couple of observations of the ClO data that need not be addressed for this manuscript but may serve as food for thought should there be a future processing version. Figure 9 is reproduced below* (**see RC1 in the interactive discussion**)*:*
*In the V8 results, there is a distinct step (increase) in the retrieved background ClO level after the instrument was switched to the reduced resolution mode (i.e., 2005 and later). There also appears to be a persistent slope to the data during background periods between polar winter events for the reduced resolution period, but there is no hint of a slope for the full resolution period. That suggests a possible artifact in the retrieval that is significant only for the reduced resolution period. There is no apparent discrepancy between full resolution and reduced resolution in the V5 results.*

[Figure]

**Figure 1.** Simulated EMAC daytime (SZA < 94°) ClO volume mixing ratios versus local time for 1–2 September, 2005, at 31.62 hPa in the latitude band 60°S–90°S. The black and red squares are MIPAS and MLS ClO VMRs averaged over daytime measurements of 1–2 September 2005–2012 in the respective latitude band, plotted against the mean local solar time of the measurements. EMAC and MLS data are convolved with a MIPAS averaging kernel. The vertical lines denote the standard errors of the mean (SEMs) of the measurements.

**Reply** At the moment, we do not have an explanation for the step in the background level and for the slope in retrieved V8 ClO of the reduced resolution period. However, both the step as well as the persistent slope are rather small, about 0.05 ppbv only. These issues will be revisited should the MIPAS data processing algorithms be updated in the future.

**RC1** *A portion of Figure 15 is reproduced above. The arrows indicate the stratospheric ClO peak seen by MLS that MIPAS never seems to fully capture. This is perhaps more evident in the difference plot between MIPAS and MLS from Figure 14, reproduced below:*

   *MIPAS is persistently lower around 2 to 3 hPa and persistently higher near 1 hPa, which suggests the ClO retrieval is smearing the peak's contribution in altitude. Since this is presumably associated with the altitude resolution of your retrieval around 1-3 hPa, I am not sure if there is anything that can be done to improve the situation, but I thought I would mention the issue in case there was. The systematic blue feature at the bottom of the above plot appears to be associated with enhanced tropical ClO in the MLS data, which I am not convinced is real, so not a problem in the MIPAS results.*

**Reply** We also think that, compared to MLS, the MIPAS ClO retrieval is smearing the peak's contribution in altitude. This assumption is confirmed by the good agreement of the MIPAS profiles with the convolved MLS profiles in Fig. 15b-e. The differences persisting after convolution in Fig. 15a and 15f are caused by the differences in local solar time. We think that we should not try to improve the situation by weakening the constraint, because the retrieval error at this altitude is already about 100% (see Fig. 6 in the manuscript) We added the sentence

"In this display it is also clearly visible that the MIPAS ClO retrieval is smearing the upper stratospheric maximum in altitude."

after the sentence "... situated at a 1–2 hPa lower pressure level." on P. 28, L. 16.

**RC1** *Minor comments:*

> *The caption to Figure 1a mentions a green dashed line, but the only panel that features a green dashed line is Figure 1b.*

**Reply** The reviewer is right. We shifted the sentence
"green dashed line: measurement noise in terms of noise equivalent spectral radiance (NESR)"
to point **(b)** of the caption.

**RC1 >** *Page 14, line 2: internal line shape (ILS) Do you not mean instrumental line shape," defined as ILS in the footnotes to Table 5?*

**Reply** The reviewer is right. We changed
"internal line shape" into "instrumental line shape."

**RC1 >***In the titles for Figures 13a, 13b, 14b, and 14c: CLO Should be ClO, without the capital "L".*

**Reply** The titles have been corrected accordingly.

**Replies to the comments of reviewer #2**:

**RC2** *Glatthor et al. is a nice manuscript introducing a new version of the 2002–2012 MIPAS ClO measurements. It is a very welcome contribution to the available data on stratospheric ClO, both in the upper stratosphere and in the lower stratosphere during the polar winter.*

**Reply** We thank reviewer #2 for this encouraging assessment.

**RC2** *Pg. 1, L. 28 – "These days, monitoring of stratospheric ClO from the ground is a routine activity". Yes, NDACC microwave measurements of ClO are "routine" in the sense that they are available on most days, but this sentence seems to imply that they are generally available at NDACC stations. There are only 2 instruments measuring ClO from the ground, one at Mauna Kea and one at Scott Base.*

**Reply** The reviewer obviously means P. 2, L. 28. We replaced the passage
"These days, monitoring of stratospheric ClO from the ground is a routine activity, in particular by stations associated with the Network for Detection of Atmospheric Composition Change (NDACC) (e.g. Solomon et al., 1984; de Zafra et al., 1994; Raffalski et al., 1998; Solomon et al., 2000; Nedoluha et al., 2011, 2025). These measurements were complemented by observations within the framework of specific measurement campaigns, using ground-based (de Zafra et al., 1989), airborne (e.g. Crewell et al., 1994; Wehr et al., 1995) and balloon-borne (e.g. Stachnik et al., 1992, 1999; Wetzel et al., 2010; de Lange et al., 2012) platforms."
by
"Since the 1980s and 1990s measurements of ClO in the microwave region from the ground have been performed at several stations, which are now associated with the Network for Detection of Atmospheric Composition Change (NDACC) (e.g. Solomon et al., 1984; de Zafra et al., 1994; Raffalski et al., 1998; Solomon et al., 2000; Nedoluha et al., 2011, 2025). These measurements were complemented by observations within the framework of specific measurement campaigns, using ground-based (de Zafra et al., 1989), airborne (e.g. Crewell et al., 1994; Wehr et al., 1995) and balloon-borne (e.g. Stachnik et al., 1992, 1999; Wetzel et al., 2010; de Lange et al., 2012) platforms. These days, routine monitoring of ClO is carried out by the NDACC microwave instruments at

Mauna Kea, Hawaii, and Scott Base, Antarctica."

**RC2** *Pg 3. L. 8 – "For data users who prefer not to work with averaging kernels, we also provide the data on a coarse grid, where averaging kernels do not need to be applied". Does this mean that there are two different retrievals being performed, or just that a smoothed version of the retrieval is supplied?*

**Reply** As outlined in Section 7.5, we performed additional coarse grid ClO retrievals for the entire MIPAS data set. To make this clearer, we changed the passage

"... we also provide the data on a coarse grid, where averaging kernels do not need to be applied."
on P. 3, L. 8 into
"... we also performed independent coarse grid retrievals, to which averaging kernels need not be applied. These retrievals will be discussed in Sect. 7.5."
and the passage
"... we offer an alternative data product where the related averaging kernels are unity;"
on P. 14, L. 16f into
"... we offer an alternative data product, obtained by a retrieval in which the related averaging kernels are unity;"

**RC2** *Pg 6., L. 13 – "The O3 results from the combined ClO–O3 retrieval are discarded because they are deemed inferior to the standard V8 O3 results." This should obviously never be the case for optimally chosen parameters for a combined ClO–O3 retrieval, but perhaps a clarification of what is meant by "inferior" would help here. Is it just the case that occasionally the inclusion of noisy channels needed for the ClO retrieval (and not the standard O3 retrieval) cause a bad O3 retrieval? Or does the inclusion of ClO adversely affect almost all O3 retrievals?*

**Reply** The $O_3$ results from the combined ClO–$O_3$ retrieval are deemed inferior to the standard V8 $O_3$ results, because these microwindows are optimized for ClO retrieval only. However, since a complete avoidance of $O_3$ signatures in the ClO microwindows is not possible, $O_3$ is jointly fitted. To make things better understandable, we changed the passage on P.6, L.9–14:

"Version 8 ozone, which is also available, is used as first guess and a priori of a combined ClO and $O_3$ retrieval. This is because $O_3$ interferences in the ClO analysis windows are so large that even minor spectroscopic inconsistencies between the lines used for the standard ozone retrieval and those used in the ClO windows could have a sizable effect on the ClO results. Further, potential weak effects of non-local thermodynamic equilibrium (non-LTE) in these $O_3$ lines are also caught by this approach. The $O_3$ results from the combined ClO–$O_3$ retrieval are discarded because they are deemed inferior to the standard V8 $O_3$ results."
into
"Because $O_3$ interferences in the ClO analysis windows are so large that even minor spectroscopic inconsistencies between the lines used for the standard V8 ozone retrieval and those in the ClO windows could have a sizable effect on the ClO results, $O_3$ is jointly fitted in the ClO retrieval. The standard V8 $O_3$ retrieval product is used as first guess and a priori. By jointly fitting of $O_3$, potentially small effects of non-local thermodynamic equilibrium (non-LTE) in these $O_3$ lines are also captured. The jointly fitted $O_3$ results are discarded, because they are deemed inferior to the standard V8 $O_3$ results obtained in microwindows dedicated for $O_3$ retrieval."

**RC2** *Figure 3 – What does the ClO profile used here look like besides being "strongly enhanced"?*

**Reply** Actually, some more information on the "strongly enhanced" ClO profile is given on P. 9, L. 29–31: "... strongly enhanced ClO (2.65 ppbv) on 4 September 2007 at 21.3 km altitude in the Antarctic lower stratosphere (73.13°S, 162.09°W, solar elevation angle 6.67°)."

For more information in the caption of Fig. 3, we changed "... strongly enhanced ClO ..." into "... strongly enhanced ClO (2.65 ppbv) ...".

**RC2** *Also, it would be very helpful here if the ClO spectrum from Figure 1b could be plotted as a third panel with the same horizontal axis as Figures 3a and 3b.*

**Reply** We added the ClO spectrum to Figure 3 and appended the sentence
"The green line in Fig. 3b indicates the spectral signature corresponding to the retrieved ClO profile and corroborates that the improvements in RMS (blue versus red) are caused by the modelled ClO." after the sentence ending in "... is visible in the red negative spikes in the residual spectrum (Fig. 3b)." on P., L. 1/2.

**RC2** *Doesn't the fact that, wherever there is a clear red/blue difference the residual is still very negative imply that there is still not enough ClO in the model?*

**Reply** The residuals are dominated by spectral noise and are not significantly more negative at the spectral positions of the ClO lines. However, the red/blue differences in the residuals show that the retrieval without ClO (red) cannot compensate for the emission in the ClO lines, which are modelled when ClO is included in the retrieval (blue). This probably becomes more obvious by comparison with the ClO spectrum added to Fig. 3, showing that the improvements in the adjustment (blue minus red spikes) are often nearly identical to the green ClO signatures.

**RC2** *Pg. 14, L. 17 – "where the related averaging kernels are unity; i.e. the profiles are free of formal a priori information". This can never be the case. "Near unity" and "contain minimal a priori information" would be okay.*

**Reply** Here we disagree with the reviewer. Unity averaging kernels can well be the case for an unconstrained retrieval free of apriori information, as performed in our coarse grid retrievals. In the ClO CGR we obtain unity averaging kernels in the height region of about 12–40 km (see Fig. 2 below). Therefore we did not change the text in response to this comment.

**RC2** *It would be good to see these kernels as a fourth panel on Figure 4, which would also show the altitude range where these coarse resolution kernels are near unity.*

**Reply** We find it unfavourable to add additional coarse grid kernels to Fig. 4, because in the associated text in Section 7.1 we discuss the averaging kernels of the standard retrieval only. Instead, we added a third contour plot to Fig. 7, showing the diagonal elements of the CGR averaging kernels along orbit 28825 (see Fig. 2 below), and discuss this plot with the sentences
"Figure 7c illustrates the diagonal elements of the averaging kernels (AKDs) of the ClO coarse grid retrieval. Except at the bottom of the retrieval range, the averaging kernels at each pressure level at and below 2 hPa have a peak value of 1."
at the end of Section 7.5 (P. 18, L. 9).

**RC2** *Pg. 31, L. 13 – The chemistry of the lower stratospheric peak and the upper stratospheric peak is completely different, so it's not clear from the data here whether the differences are larger because of a larger local time difference or to differences in the sensitivity to local time.*

**Reply** We do not discuss the differences between the lower and the upper stratospheric peaks here. In any case, the differences in local solar time of MIPAS and MLS measurements are larger at polar latitudes than at the Equator. For this reason we think that it is rather plausible that the differences between measurements of a tracer exhibiting a diurnal variation become larger at higher latitudes.

[Figure]

**Figure 2.** Diagonal elements of the averaging kernels of the V8 ClO coarse grid retrieval along orbit 28825.

We did not change the text in response to this comment.

**RC2** *More generally, it would be very helpful somewhere earlier in the manuscript (perhaps on Pg. 26, if not earlier) to discuss the local times of both the MLS and MIPAS measurements, rather than to have the reader wondering about the cause of the bias between MLS and MIPAS shown in Figure 14. Currently it is not until the final figure (Figure 18) of the paper that the sensitivity of daytime measurements to local time is discussed.*

**Reply** The local solar times of the Equatorial crossing of MIPAS and MLS measurements are already specified in Section 2, and the reason for the increasing differences in local solar time are discussed on P. 31, L. 13–16. For more clarity earlier in the manuscript we added the passage
"Because of the opposite flight directions and the turn of the viewing direction of MIPAS towards the poles at latitudes higher than 80°, the differences in local solar time between MIPAS and MLS measurements at high latitudes become even larger, especially above the Antarctic."
after P. 3, L. 31 in Section 2.

**RC2** *Pg. 33, last line – "the MIPAS and MLS data observed during Antarctic winter perfectly fit to the EMAC curve". Since ClO in this region is dependent on the details of the presence of PSCs and therefore, when averaged over a large latitude band, on the precise variations in local temperature, the "perfect fit" in Figure 18a for this particular date seems serendipitous. Are the fits on other days during this period similarly good?*

**Reply** Actually we state a nearly perfect agreement for the data from 1–2 July at 3.16 hPa shown in Fig. 18d. At this pressure level PSCs do not occur. Unfortunately we do not have EMAC data for other days closely around 1–2 July. However, on request of reviewer #1 we performed additional EMAC model calculations for 1–2 September 2005. For this period, the Antarctic MIPAS and MLS ClO VMRs at 3.16 hPa also fit well to the EMAC curve (see Fig. 3 below). We added the sentence
"Further, these additional model calculations show that the smaller bias between Antarctic MIPAS and MLS ClO at 3.16 hPa during September is also well explained by the difference in local solar time."
after the additions at the end of Section 9.4 (P. 33, L. 2) outlined in our response to reviewer #1.

[Figure]

**Figure 3.** Simulated EMAC daytime (SZA < 94°) ClO volume mixing ratios versus local time for 1–2 September, 2005, at 3.16 hPa in the latitude band 60°S–90°S. The black and red squares are MIPAS and MLS ClO VMRs averaged over daytime measurements of 1–2 September 2005–2012 in the respective latitude band, plotted against the mean local solar time of the measurements. EMAC and MLS data are convolved with a MIPAS averaging kernel. The vertical lines denote the standard errors of the mean (SEMs) of the measurements.